# SLAMF7 and IL-6R define distinct cytotoxic versus helper memory CD8+ T cells

Lucie Loyal[1,2], Sarah Warth[2,3], Karsten Jürchott[2,4,5], Felix Mölder [6], Christos Nikolaou [2,7], Nina Babel[8], Mikalai Nienen[5], Sibel Durlanik[2], Regina Stark [9], Beate Kruse[1,2], Marco Frentsch[2], Robert Sabat[7], Kerstin Wolk [7] & Andreas Thiel [1,2 ✉]

The prevailing 'division of labor' concept in cellular immunity is that CD8+ T cells primarily utilize cytotoxic functions to kill target cells, while CD4+ T cells exert helper/inducer functions. Multiple subsets of CD4+ memory T cells have been characterized by distinct chemokine receptor expression. Here, we demonstrate that analogous CD8+ memory T-cell subsets exist, characterized by identical chemokine receptor expression signatures and controlled by similar generic programs. Among them, Tc2, Tc17 and Tc22 cells, in contrast to Tc1 and Tc17 + 1 cells, express IL-6R but not SLAMF7, completely lack cytotoxicity and instead display helper functions including CD40L expression. CD8+ helper T cells exhibit a unique TCR repertoire, express genes related to skin resident memory T cells (T$_{RM}$) and are altered in the inflammatory skin disease psoriasis. Our findings reveal that the conventional view of CD4+ and CD8+ T cell capabilities and functions in human health and disease needs to be revised.

[1] Si-M/"Der Simulierte Mensch" a science framework of Technische Universität Berlin and Charité-Universitätsmedizin Berlin, 13353 Berlin, Germany. [2] Regenerative Immunology and Aging, BIH Center for Regenerative Therapies, Charité-Universitätsmedizin Berlin, 13353 Berlin, Germany. [3] BCRT Flow Cytometry Lab, Berlin-Brandenburg Center for Regenerative Therapies (BCRT), Charité-Universitätsmedizin Berlin, 13353 Berlin, Germany. [4] Applied Bioinformatics, Berlin-Brandenburg Center for Regenerative Therapies (BCRT), Charité-Universitätsmedizin Berlin, 13353 Berlin, Germany. [5] Institute for Medical Immunology, Charité-Universitätsmedizin Berlin, 13353 Berlin, Germany. [6] Genome Informatics, Institute of Human Genetics, Universität Duisburg-Essen, 45147 Essen, Germany. [7] Psoriasis Research and Treatment Center, Dermatology/Medical Immunology, Charité-Universitätsmedizin Berlin, 10117 Berlin, Germany. [8] Medical Clinic I, Marien Hospital Herne, Ruhr Universität Bochum, 44625 Herne, Germany. [9] Adaptive Immunity Lab, Department of Hematopoiesis, Sanquin Blood Supply Foundation, 1066 Amsterdam, The Netherlands. ✉email: andreas.thiel@charite.de

T cells circulate through the body or reside in tissues pending pathogen challenge. The classical "division of labor" concept of adaptive cellular immunity separates cytotoxic CD8+ T cells with the capacity to kill infected cells from CD4+ T cells that express CD40L, and provide help to APCs and B cells. In recent years, it has become evident that T cells are additionally equipped with a vast functional diversity to provide pathogen-tailored responses[1–3]. During the activation of naive (N) T cells, cytokine milieu-induced transcriptional programs result in the formation of distinct memory T-cell subsets with unique functions and cytokine profiles[4]. Among CD4+ memory T cells, these cells can be identified and distinguished ex vivo based on the expression patterns of chemokine receptors related to their homing potential. Combinations of the four chemokine receptors CCR4, CCR6, CCR10, and CXCR3 were utilized for the identification of Th1, Th17 + 1, Th2, Th17, and Th22-type CD4+ memory T-cell subsets[5–8]. Despite the thoroughly assessed CD4+ memory T-cell heterogeneity, much less is known about the diversity of memory CD8+ T cells. The strategy to delineate functional subsets among CD4+ T cells, using distinct chemokine receptor expression patterns has been only partly adapted to CD8+ T cells. CXCR3 and CCR4 were used for the identification of Tc1 and Tc2 CD8+ T cells that produce IFN-γ and IL-4/IL-13 analogous to the corresponding CD4+ T-cell subsets[9]. Also IL-17-producing CCR6+ CD8+ T cells were reported[10]. However, available data so far reported heterogeneous functional profiles for such cells producing either IL-17 alone or in combination with IL-22, IL-10, or IFN-γ[11]. Reports were also inconsistent with respect to the cytotoxic capacity of different memory CD8+ T cells, especially those associated with a Tc2 or Tc17 phenotype[11–13]. We have previously described a prominent fraction of CD8+ memory T cells lacking cytotoxic features and instead expressing the helper molecule CD40L, and being capable of exerting helper-type functions in vitro and in vivo[14]. While the differentiation of CD4+ T cells into cytotoxic cells has been characterized[15,16], it has remained unclear whether and how the formation of distinct cytotoxic versus non-cytotoxic functions in memory CD8+ T cells is regulated. In order to gain a greater understanding of memory CD8+ T-cell compartment diversification, we therefore conduct a systematical and detailed analysis of human circulating CD8+ memory T-cell subsets, and compare them to well-characterized CD4+ memory T-cell subsets. Analogous to CD4+ memory subsets, chemokine receptor expression signatures enable the delineation of different CD8+ memory T-cell subsets Tc1, Tc2, and Tc17, but also Tc17 + 1 and Tc22 with complementary cytokine expression profiles. Intriguingly, Tc2, Tc17, and Tc22 cell subsets, in contrast to Tc1 and Tc17 + 1 cells, completely lack the features of cytotoxic cells and instead display helper features, including the ability to express CD40L. CD8+ helper T cells express IL-6R, but not SLAMF7, a molecule expressed by all cytotoxic lymphocytes and exhibit a unique TCR repertoire. Our findings support the notion that memory CD8+ T cells are as multifunctional as memory CD4+ T cells. The here classified distinct subsets of CD8+ helper T cells may be pivotal in protecting barrier sites, such as the skin, but their deregulation can also contribute to the pathogenesis of autoimmune and allergic inflammatory disorders. Moreover, our data demonstrates that cytotoxic and helper phenotypes among memory T cells represent generic immunological codes independent of their MHC-I or MHC-II restriction, as well as associated CD4 versus CD8 expression.

## Results

### Chemokine receptors delineate CD4+ and CD8+ T-cell subsets.
We first assessed the expression of human CD4+ memory T-cell subsets defining chemokine receptors CXCR3, CCR4, CCR6, and CCR10 in circulating human CD8+ memory T cells (Supplementary Fig. 1a). In the peripheral blood of healthy donors, CXCR3 was expressed among CD45RA+ CCR7+ N, CD45RA− CCR7+ central memory (CM), CD45RA− CCR7− effector memory (EM), and CD45RA+ CCR7− EM with reacquired CD45RA expression (EMRA) CD8+ T cells. In contrast, the expression of CCR4 and CCR6 was almost exclusively restricted to EM and CM CD8+ T cells, and only CCR10 was additionally found in the fraction of EMRA CD8+ T cells. As previously described, combinations of the chemokine receptors delineates the CD45RA− CD4+ memory T-cell compartment into Th1 (CXCR3+CCR4−CCR6−), Th2 (CXCR3−CCR4+CCR6−), Th17 + 1 (CXCR3+CCR4−CCR6+), Th17 (CXCR3−CCR4+CCR6+CCR10−), and Th22 (CXCR3+CCR4+CCR6+CCR10+) cells according to their functional capacities (Fig. 1a). We could demonstrate the presence of CD8+ T cells with Tc1- and Tc2-type signatures, but also signatures correlating with Th17 + 1, Th17, and Th22 cells (here referred to as Tc17 + 1, Tc17, and Tc22; Fig. 1a). While the frequencies of the functionally distinct CD4+ T-cell subsets spanned between 5 and 20% of CD45RA− memory CD4+ cells, among memory CD8+ T cells, Tc1 cells dominated with an average frequency of 38% followed by Tc17 + 1 cells (10%), Tc2 cells (8%), and Tc17 and Tc22 cells (~1% each) in healthy donors (Fig. 1b). Tc2, Tc17, and Tc22 cells were rather found among CCR7+ CM CD8+ T cells, whereas Tc1 and Tc17 + 1 CD8+ T cells more often display a CCR7− EM phenotype (Supplementary Fig. 1b). The flow cytometric cytokine secretion profiles of ex vivo stimulated, sorted Tc-cell subsets (Fig. 1c), as well as analysis of cell supernatants from 24 h and 72 h after stimulation confirmed the classification into the different functional subsets (Fig. 1d and Supplementary Fig. 1c, d). In accordance with their CD4+ Th-cell counterparts, Tc-cell subsets produced similar distinct sets of key effector cytokines.

### T-cell subsets utilize equal sets of differentiation programs.
To further characterize the relation of distinct CD4+ and corresponding CD8+ memory T-cell subsets, we compared transcriptomes of sorted CD4+ and analogous CD8+ T-cell subsets by whole RNA sequencing. The heatmap of the top 1000 most differentially expressed genes among all Tc- and Th-cell subsets, as well as the principal component analysis (PCA) covering all protein-coding genes demonstrated a strong correlation in the gene expression signatures between each CD8+, and its respective CD4+ T-cell subset upon batch analysis (Fig. 2a, b). We confirmed these associations in an unbiased way by the identification of differentially expressed genes among the Th-cell subsets of published human and mouse GEO datasets, and analyzed their overlap with the gene expression pattern of our Tc-cell subsets (Supplementary Fig. 2a). The similarity between the CD8+ and their respective CD4+ T-cell subsets was further emphasized by the expression of corresponding key transcription factors that are essential for the differentiation and maintenance of individual Th-cell subsets (Fig. 2c). While the expression of TBX21 was restricted to Th1/Tc1 and Th17 + 1/Tc17 + 1, and RORC to Th17 + 1/Tc17 + 1 and Th17/Tc17 cell subsets, GATA3 gene expression and protein was detectable in Th2, Tc2, but also in Tc17 and Tc22 cell subsets (Fig. 2c, d). In contrast, all CD4+ T-cell subsets as well as Tc2, Tc17, and Tc22 cell subsets expressed AHR (Fig. 2c, d). Thus, analogous differentiation patterns in corresponding CD4+, as well as CD8+ memory T-cell subsets are determined by similar generic molecular codes.

### Tc2, Tc17, and Tc22 T-cell subsets possess helper characteristics.
Since the unsupervised clustering of normalized data separated the CD8+ Tc1 and Tc17 + 1 cell subsets from the CD8+

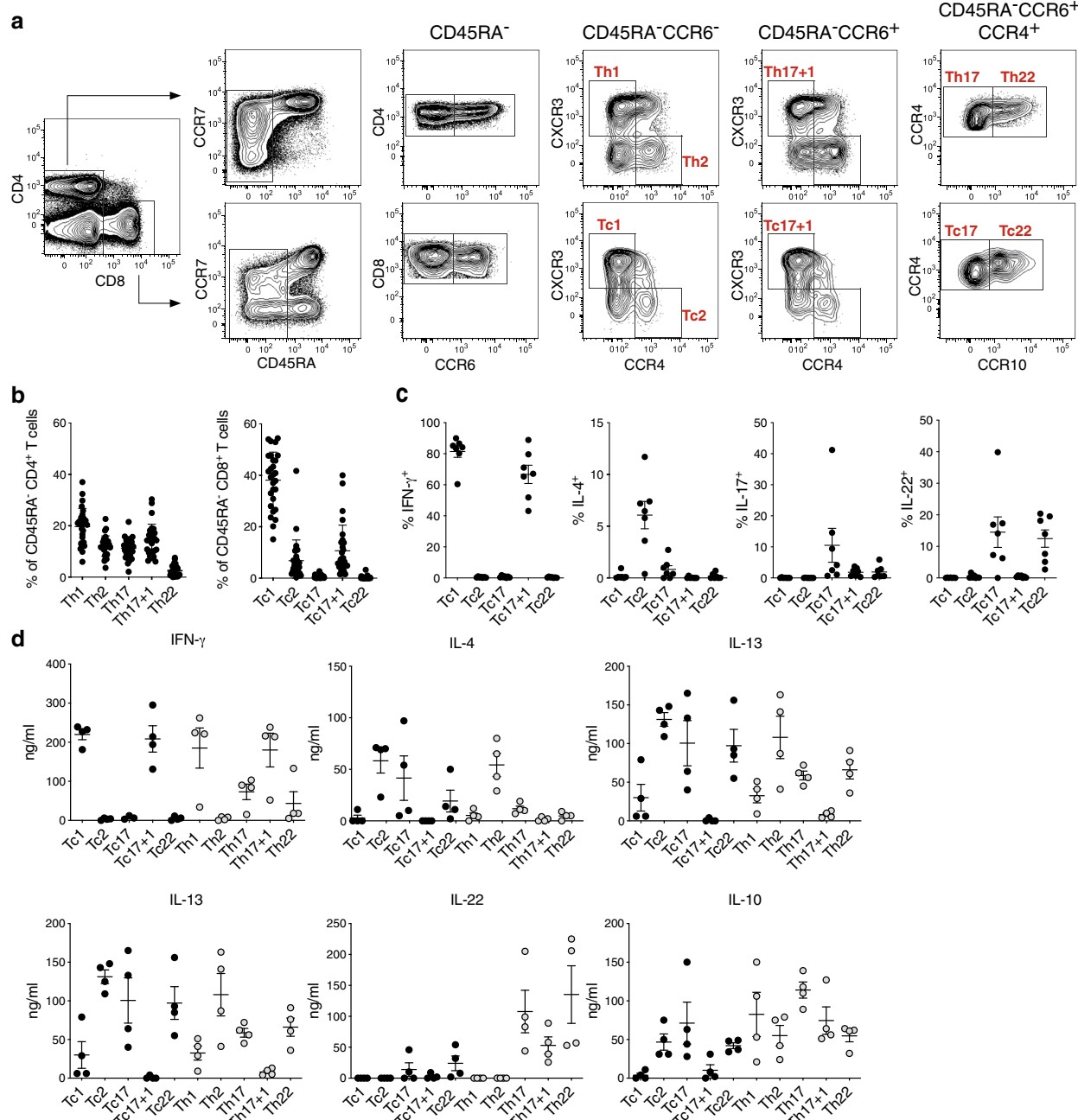

**Fig. 1 Chemokine receptors identify CD4$^+$ and corresponding CD8$^+$ memory T-cell subsets. a** Representative FACS plot of chemokine receptor-based gating strategy in PBMC. Th/Tc subsets were gated within CD45RA$^-$CD4$^+$ or CD8$^+$ memory T cells as following: Th1/Tc1 CCR6$^-$CXCR3$^+$CCR4$^-$, Th2/Tc2 CCR6$^-$CXCR3$^-$CCR4$^+$, Th17 + 1/Tc17 + 1 CCR6$^+$CXCR3$^+$CCR4$^-$, Th17/Tc17 CCR6$^+$CXCR3$^-$CCR4$^+$CCR10$^-$, and Th22/Tc22 CCR6$^+$CXCR3$^-$CCR4$^+$CCR10$^+$. **b** Percentage of CD45RA$^-$ CD4$^+$ and CD8$^+$ T cells expressing the defined chemokine receptor pattern for the specific Th- or Tc-cell subsets respectively ($n = 28$), mean ± SEM. **c** Frequencies of cytokine secreting cells among sorted and polyclonally activated Tc-cell subsets ($n = 7$), mean ± SEM. **d** Sorted Tc- (black circles) and Th- (gray circles) cell subsets were stimulated for 72 h and the cytokine concentrations of the supernatants assayed by Multiplex ELISA ($n = 4$ for each subset), mean ± SEM.

Tc2, Tc17, and Tc22 cell subsets, and clustered the latter ones with the CD4$^+$ T-cell subsets, we assumed that they might represent a CD8$^+$ T-cell fraction with CD4$^+$ helper T-cell characteristics (Supplementary Fig. 2b). Gene ontology overrepresentation analysis (ORA) displayed a high enrichment of immunoregulatory genes and genes that are indicative of tissue migration in helper-type Tc2, Tc17, and Tc22 cells, while only Tc1 and Tc17 + 1 cells expressed genes associated with cytotoxicity, such as *Granzymes*, *Perforin*, *CRTAM*, and *NKG7* (Fig. 3a and Supplementary Fig. 2c).

Despite their striking similarity, Tc1 and Tc17 + 1 cells clearly differ from each other in the expression of several genes, such as *IL23R*, *IFNGR1*, *VCAM1*, *CD38*, *TIGIT*, and *CCR1*, while Tc2, Tc17, and Tc22 cells share different combinations of genes, including *PTGDR2* (CRTH2), *ICOS*, *TNFSF11* (RANK), and *ITGAE* (CD103; Fig. 3a). Flow cytometric analysis identified perforin and granzyme B in resting Tc1 and Tc17 + 1 cells, while Tc2, Tc17, and Tc22 cells expressed the skin-homing marker CLA instead (Fig. 3b, c). Upon activation, Tc2, Tc17, and Tc22 cells

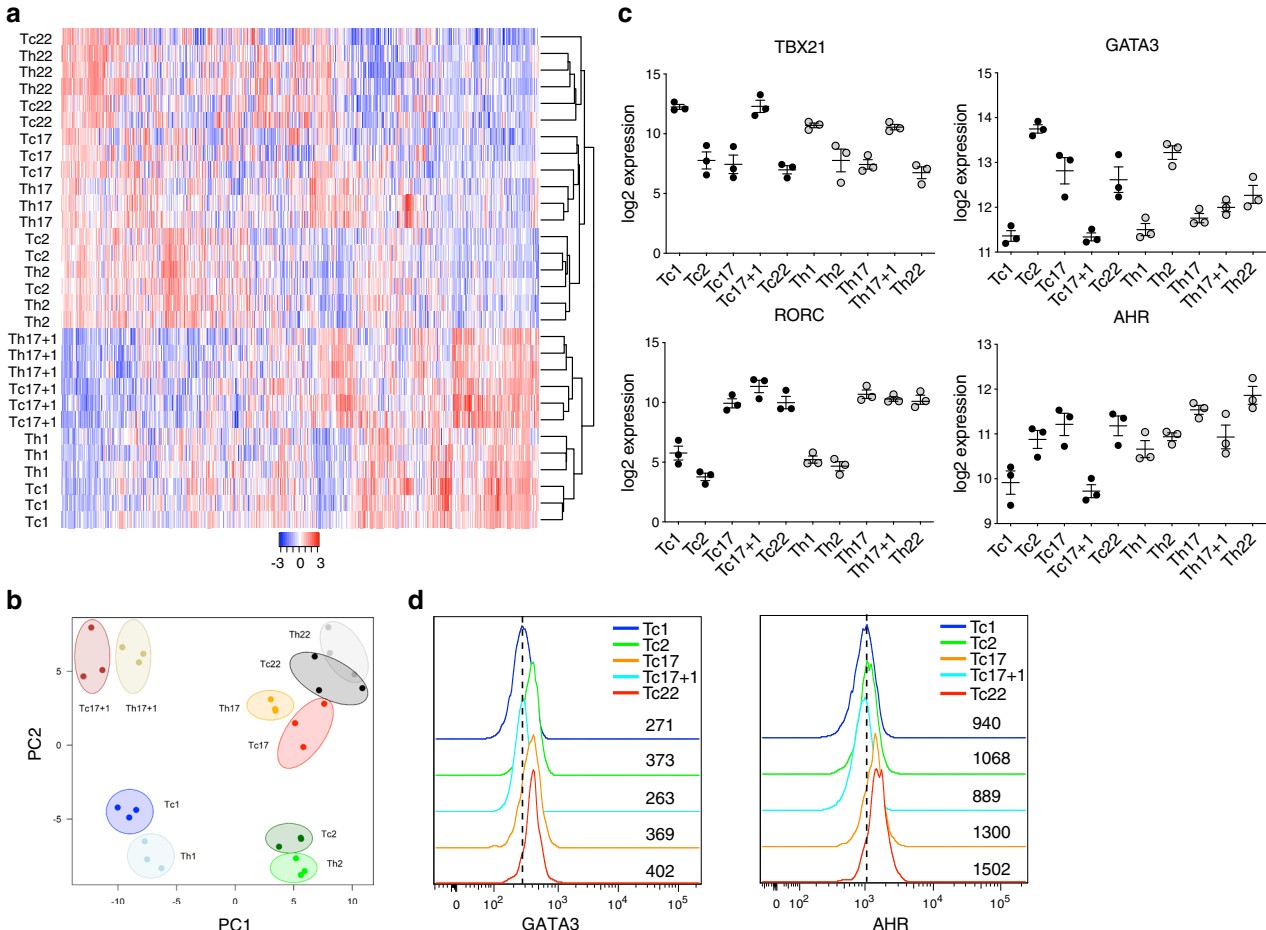

**Fig. 2 CD4$^+$ Th- and respective CD8$^+$ Tc-cell subsets utilize the same differentiation programs. a** Heatmap of the top 1000 most variable expressed genes of sorted CD8$^+$ and CD4$^+$ memory T-cell subsets as shown in Fig. 1a after normalization and batch compensation (number of donors = 3). **b** Principal component analysis of all protein-coding expressed genes. **c** Normalized, log2-transformed expression of indicated transcription factors among Tc- (black circles) and Th- (gray circles) cell subsets, mean ± SEM. **d** Representative histogram of intracellular staining for transcription factors GATA3 and AHR in CD8$^+$CD45RA$^-$ Tc-cell subsets. Numbers in plots indicate the MFI.

homogenously expressed the helper molecule CD40L at intensities comparable to CD4$^+$ T cells (Fig. 3d and Supplementary Fig. 2d). In line with these results, the secretion of IL-17A and IL-13 was limited to CD40L$^+$, but not CD40L$^-$ CD8$^+$ memory T cells in immune competent pet shop mice, while such cells were barely detectable in specific pathogen-free (SPF) mice (Fig. 3e and Supplementary Fig. 2e). Altogether, Tc2, Tc17, and Tc22 cells comprise a unique CD8$^+$ memory T-cell fraction with helper and immunomodulatory features rather than classical cytotoxic capacities.

**Helper CD8$^+$ memory T cells contribute to skin immunity.** Since one of the most prominent differences distinguishing cytotoxic CD8$^+$ Tc1 and Tc17 + 1 cells from helper Tc2, Tc17, and Tc22 cells was the expression of skin-homing markers, we analyzed how CD8$^+$ helper T-cell subsets are related to CD8$^+$ skin T$_{RM}$ and non-T$_{RM}$ cells. In silico comparison of CD8$^+$ T-cell subsets gene expression signatures with published data from various skin-derived epidermal T$_{RM}$, dermal non-T$_{RM}$, and blood-derived T$_{EM}$-cell subsets as control[17], placed CD8$^+$ helper T cells in the proximity of CD8$^+$ T cells isolated from skin, while circulating CLA$^+$/CLA$^-$ T$_{EM}$, as well as Tc1/Tc17 + 1 cell subsets formed separate clusters (Fig. 4a). Moreover, utilizing mass cytometry, we detected the expression of the T$_{RM}$ marker CD103 among Tc17/Tc22 cells, whereas CD69 and CD49a (refs. [17,18]) were absent in all other circulating CD8$^+$ T-cell subsets analyzed (Fig. 4b and

Supplementary Fig. 3a). Accordingly, the T$_{RM}$-associated markers CD101 and CD9 were highly expressed in these subsets (Fig. 3b)[19]. Finally, we assessed a contribution of CD8$^+$ helper T cells in psoriasis, a prototype inflammatory skin disease. In the peripheral blood of psoriasis patients, Tc2, Tc17, and Tc22 CD8$^+$ helper T-cell subsets were significantly enriched, while no such changes could be observed in control atopic dermatitis patients (Fig. 4c and Supplementary Fig. 4a, b). We also revealed that activated CD8$^+$ helper T-cell subsets induced CCL20 in keratinocytes, which was strongly enriched in psoriatic, but not atopic dermatitis lesions (Fig. 4d, e). The CCL20 expression in psoriatic lesions correlated with CCR4 expression, a shared characteristics of all CD8$^+$ helper T-cell subsets (Supplementary Fig. 4c). Interestingly, all CD8$^+$ helper T-cell subsets also shared the ability to produce the cytokine IL-13 (Fig. 1d), a tissue homeostasis supporting cytokine that induces CCL26 secretion by keratinocytes, which antagonistically favors the influx of helper-type CCR4$^+$ cells above T cells with cytotoxic capacity (Fig. 4e)[20,21]. Thus, CD8$^+$ helper T cells may play a pivotal role in skin tissue immunology, and their dysregulation may contribute to the etiopathology of inflammatory skin diseases.

**CD8$^+$ T-cell subsets differ in their TCR repertoire.** Previously, Becattini et al. demonstrated that single pathogen-specific CD4$^+$ T cells can differentiate into functionally distinctive memory T-cell clones[22]. To assess whether such plasticity is also the basis

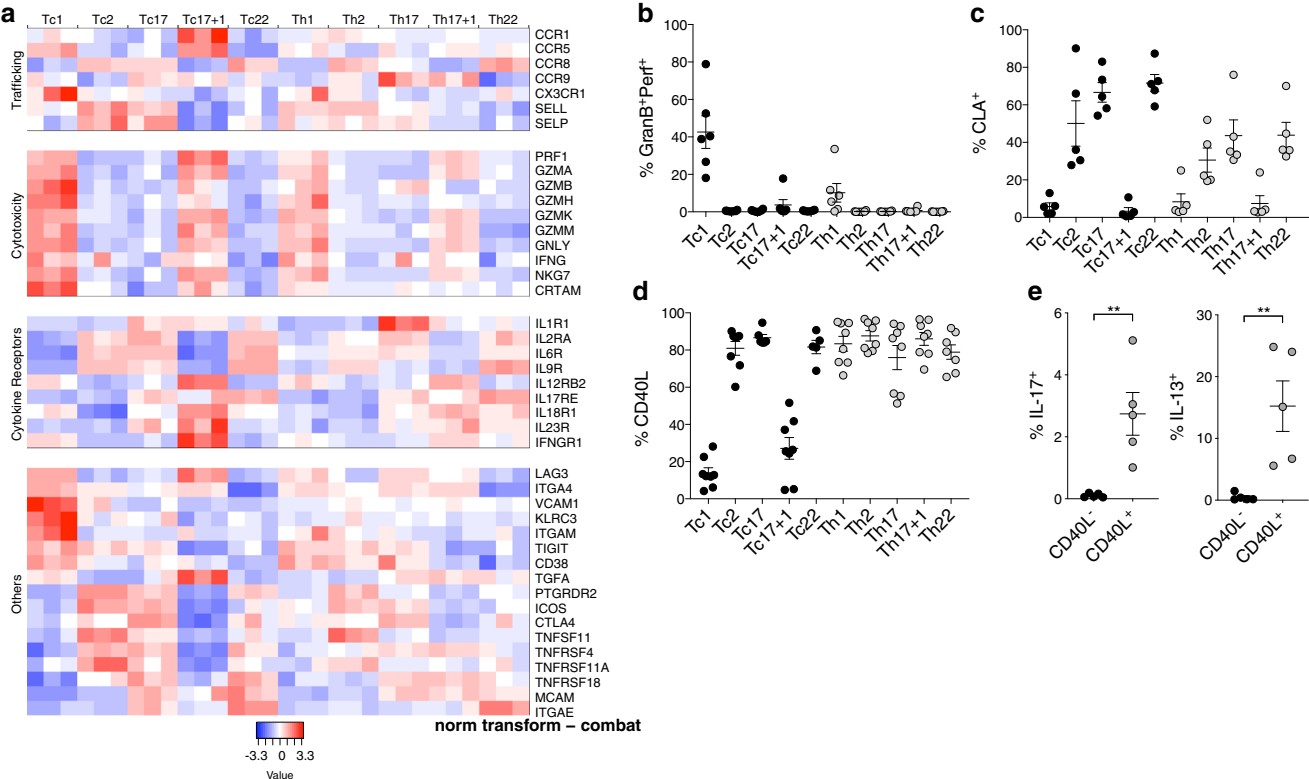

**Fig. 3 Tc2, Tc17, and Tc22 CD8$^+$ T-cell subsets possess helper characteristics. a** Heatmap representation of normalized, log2-transformed RNA-Seq expression values of selected genes. **b, c** Perforin and granzyme B (**b**) or CLA (**c**)-expressing cells among resting CD45RA$^-$CD8$^+$ (black circles) and CD4$^+$ (gray circles) T-cell subsets ($n = 6$), mean ± SEM. **d** CD40L-expressing cells among sorted and polyclonally activated Tc- (black circles) and Th- (gray circles) cell subsets ($n = 8$), mean ± SEM. **e** Frequencies of IL-17$^+$ and IL-13$^+$ cells among splenocyte derived CD40L$^-$ (black circles) or CD40L$^+$ (gray circles) dump $^-$CD3$^+$CD44$^+$CD8$^+$ memory T cells from pet shop mice after polyclonal activation. Mean ± SEM. Student's t test. *$P < 0.05$, **$P < 0.01$, ***$P < 0.001$.

for the heterogeneous CD8$^+$ memory T-cell subsets described here, we performed TCRβ-chain deep sequencing of highly purified CD8$^+$ memory T-cell subsets. An oligoclonal repertoire in Tc17 and Tc22 cells with a few dominating clones suggested a highly specialized response, in contrast to the more polyclonal Tc1, Tc17 + 1, and Tc2 cells (Fig. 5a, b and Supplementary Fig. 5a–c). The distinct Vβ-families of clones between the Tc-cell subsets of the same donor and between the donors imply that those cells are not semi-invariant CD1b-specific cells that rely on the usage of Vβ4-1. However, we observed an enrichment of Vβ13 (TRBV6) and Vβ2 (TRBV20) families among Tc17 + 1 cells, of which up to 10% could be characterized as MAIT cells based on their co-expression of Vα7.2 and CD161 (Supplementary Fig. 6)$^{23,24}$. A strong overlap of TCRβ-CDR3 clonotypes was demonstrated for the two cytotoxic Tc1 and Tc17 + 1 cell subsets and the three helper Tc2, Tc17, and Tc22 cell subsets covering ~50% of total reads (Fig. 5c, d). Therefore, given the small overlap in their shared clonotypes and shared reads (Fig. 5e), cytotoxic CD8$^+$ T cells and CD8$^+$ helper T cells are most probably involved in diverse antigenic responses. Within such responses, heterogeneous differentiation or plasticity may then allow the differentiation either of Tc1 and Tc17 + 1 cells, or of Tc2, Tc17, and Tc22 cells from single precursor clones.

**SLAMF7 and IL-6R distinguish cytotoxic and helper memory T cells.** To identify markers that can distinguish cytotoxic from helper memory T-cell subsets, we screened for genes that were differentially expressed in T-cell populations with a cytotoxic phenotype (Tc1 and Tc17 + 1) compared to all other Th- and

Tc-cell subsets, leading to the identification of 28 genes that were more weakly, and 22 genes that were more strongly expressed in cytotoxic T cells (Fig. 6a). Among them, the two cell surface molecules SLAMF7 and IL-6R displayed a striking converse expression pattern (Fig. 6b). The majority of CD4$^+$ T cells, as well as CD8$^+$ Tc2, Tc17, and Tc22 helper T-cell subsets, but not the cytotoxic Tc1 and Tc17 + 1 cell subsets, expressed high levels of IL-6R (Fig. 6c and Supplementary Fig. 7a). Opposing expression patterns were observed for SLAMF7 that was only expressed by the cytotoxic Tc1 and Tc17 + 1 CD8$^+$ memory T-cell subsets, T$_{EMRA}$, and all granzyme B and/or perforin-expressing CD8$^+$ T cells (Fig. 6d, e and Supplementary Fig. 7a). Analogously, granzyme B and perforin-expressing cytotoxic CD4$^+$ T cells constituted the small SLAMF7$^+$ T-cell fraction among CD4$^+$ T cells (Fig. 6e). In addition, further subsets with cytotoxic characteristics, such as CD56$^+$ NK, NKT, and ILC1 cells expressed SLAMF7, but not IL-6R (Supplementary Fig. 7b). We observed cell-directed killing only mediated by SLAMF7$^+$, but not IL-6R$^+$ memory CD8$^+$ and CD4$^+$ T cells (Fig. 6f and Supplementary Fig. 7c). SLAMF7$^+$ cytotoxic but not SLAMF7$^-$ helper memory CD8$^+$ T cells dominated antiviral cellular immunity against CMV, whereas ~20% of EBV or influenza M-specific CD8$^+$ T cells are SLAMF7$^-$ consisting of both N and CM T cells (Fig. 6g). In line with these findings, Tc1 cells dominated in organs, such as tonsils and lung, whereas Tc2 cells were overrepresented in the bone marrow (Fig. 6h). The helper versus cytotoxic phenotype was stable under homeostatic conditions, as well as upon stimulation and cultivation in the presence of the cytotoxicity-inducing cytokines IL-2 and IL-12 (Supplementary Fig. 7d). When assessing the usability of SLAMF7 as cytotoxic

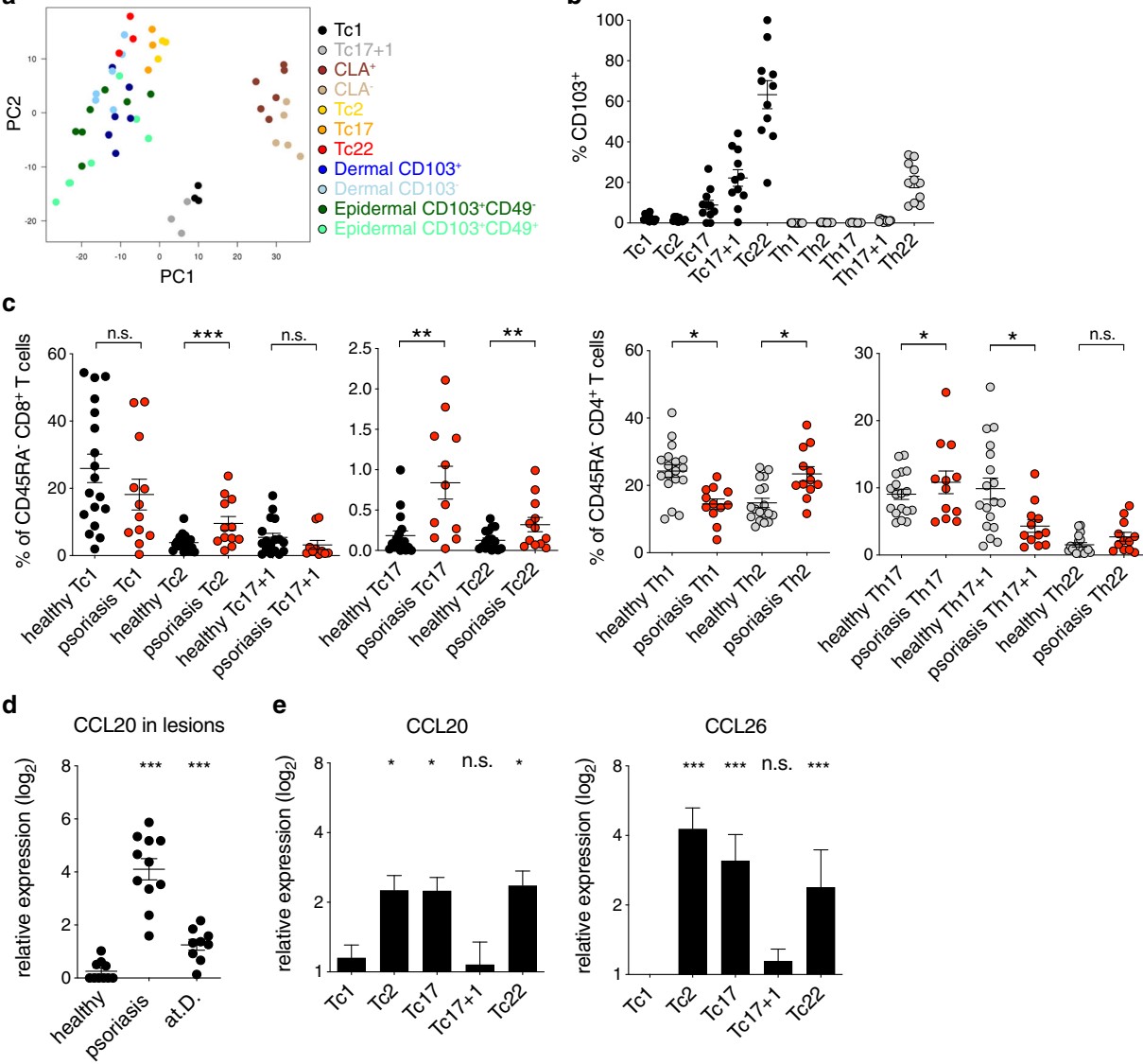

**Fig. 4 Helper CD8+ memory T cells contribute to skin immunity. a** PCA of CD8+ memory T-cell subsets and CLA+/CLA− blood-derived memory, CD103+/CD103− dermis derived, and CD49a+/CD49a− epidermis-derived CD8+ T cells of the GEO dataset GSE83637. **b** Frequencies of CD103+ T cells among indicated Tc- (black circles) and Th- (gray circles) cell subsets in human PBMCs. Mean ± SEM. **c** Frequencies of peripheral Tc- (black circles) and Th- (gray circles) cell subsets of healthy donors or psoriasis patients (red circles). Mean ± SEM. Student's t test. *P < 0.05, **P < 0.01, ***P < 0.001. **d** mRNA levels of CCL20 in healthy skin, and lesions of psoriasis and atopic dermatitis (at.D.) patients relative to healthy control (n = 10). Mean ± SEM. Student's t test. *P < 0.05, **P < 0.01, ***P < 0.001. **e** Relative expression of CCL20 and CCL26 by keratinocytes stimulated with supernatants derived from sorted and activated Tc-cell subsets (n = 5). Mean ± SEM. Student's t test. *P < 0.05, **P < 0.01, ***P < 0.001 relative to Tc1.

marker in mouse, we observed an impaired development of cytotoxic memory CD8+ T cells in SPF mice in analogy to their reduced CD8+ helper T-cell compartment. In contrast, immune experienced pet shop mice exhibit higher frequencies of perforin+ cells, and allowed a separation of cytotoxic and non-cytotoxic cells by IL-6R and SLAMF7 among murine memory CD8+ T cells (Supplementary Fig. 7e, f). A cytotoxic phenotype, including perforin expression, was associated with RUNX3 expression in CD8+, but also in CD4+ T cells[15,16,25–27]. Human cytotoxic CD8+ memory (Tc1/Tc1 + 1), cytotoxic CD4+, and cytotoxic EMRA CD8+ T cells expressed comparable levels of RUNX3, whereas all CD8+ memory helper T-cell subsets (Tc2, Tc17, and Tc22) expressed low levels of RUNX3 similar to CD4+ memory helper T cells (Fig. 6i). RUNX3 was also reported to interfere with CD40L expression[25,28]. In line with this, cytotoxic CD4+ T cells

are incapable to express CD40L upon stimulation (Fig. 6j). Hence, cytotoxicity is associated with RUNX3 expression independent of the T-cell lineage affiliation, and mutually exclusive SLAMF7 versus IL-6R expression pattern delineates resting cytotoxic and noncytotoxic lymphocytes in humans and mice.

## Discussion

Distinctive cytokine milieus regulate the differentiation of CD4+ memory T cells, including the induction of specific cytokine secretion profiles and distinctive chemokine receptor expression patterns needed for the homing to various tissues[4]. We demonstrate here that similar mechanism apply for CD8+ memory T cells and the same chemokine receptor expression patterns analogously delineate functionally distinct CD8+ memory T-cell subsets. Tc1 and Tc17 + 1 cells possess typical CD8+ T cell-related cytotoxic

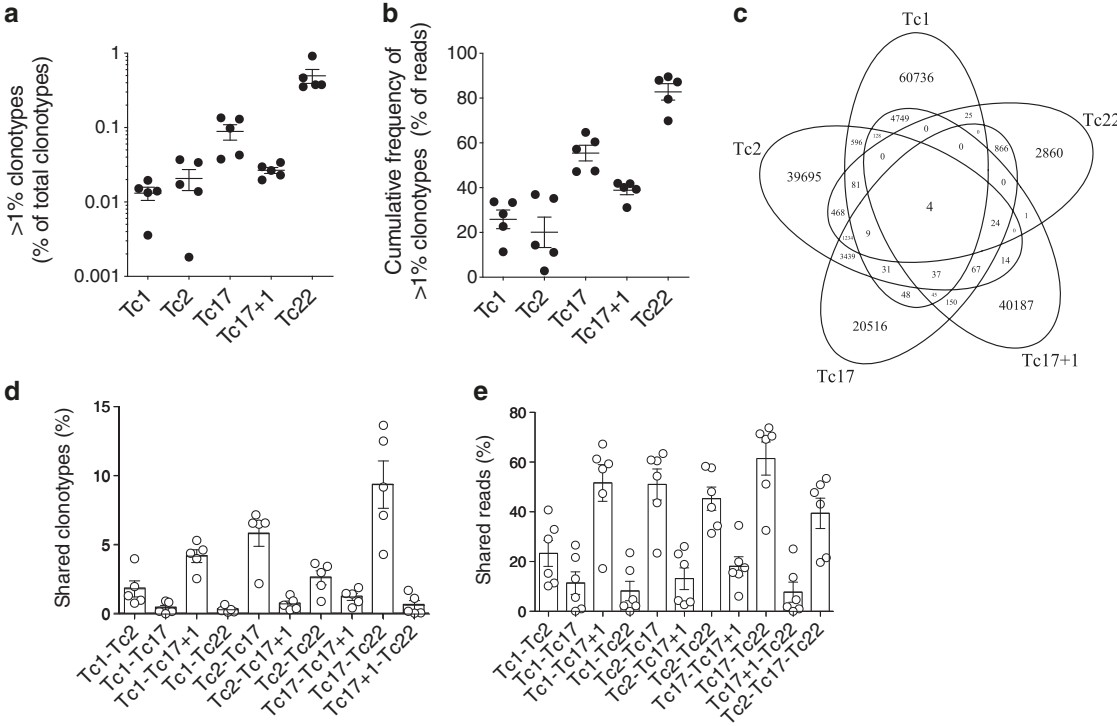

**Fig. 5 Cytotoxic and helper CD8$^+$ T-cell subsets differ in their TCR repertoire. a** Contribution of clonotypes present at frequencies >1% of reads to the total clonotype repertoire ($n = 5$), mean ± SEM. **b** Cumulative contributions of clonotypes present at frequencies >1% to the total reads in all five donors. Mean ± SEM. **c** Venn diagram showing a representative overlap of clonotypes of different Tc-cell subsets. **d, e** Bar diagram showing the percentage of shared clonotypes (**d**) and shared reads (**e**) of indicated subsets in all five donors. Mean ± SEM.

signatures accompanied by a stable expression of the CD8$^+$ lineage transcription factor RUNX3. In contrast, Tc2, Tc17, and Tc22 cells were non-cytotoxic, CD40L$^+$, expressed reduced levels of RUNX3 and displayed a shift from T-bet toward GATA3 expression. RUNX3 is a central upstream transcription factor for genes associated with cytotoxicity, such as perforin and granzyme B, and can be suppressed by GATA3[29,30]. Accordingly, we observed absent GATA3 expression in Tc1 and Tc17 + 1, but high expression in all helper-type Tc2, Tc17, and Tc22 memory T-cell subsets. We identified SLAMF7 and IL-6R as universal markers to delineate cytotoxic from helper T cells independent of their CD4 or CD8 expression. Self-ligation of SLAMF7, previously described as a NK cell marker, was recently shown to increase cytotoxic degranulation and IFN-γ secretion capacity not only in NK cells, but also in CD8$^+$ T cells[31]. We could demonstrate that antiviral CD8$^+$ T cells are mostly SLAMF7$^+$. In contrast, non-cytotoxic T cells displayed IL-6R expression allowing classical IL-6 signaling, which has been associated with anti-inflammatory processes[32]. Most strikingly, the here identified gene expression signatures support the notion that memory T-cell compartmentalization into cytotoxic versus helper-type cells, as well as different Th/Tc subsets underlie generic codes, that are shared by CD4$^+$ and CD8$^+$ T cells.

While CD4$^+$ helper T cells and cytotoxic CD8$^+$ T cells have been studied thoroughly, reports on non-cytotoxic, helper-type memory CD8$^+$ T cells remained rare, until recently. In human skin, a prominent, non-cytotoxic IL-17$^+$ CD8$^+$ T-cell population was found among epidermal T$_{RM}$ cells[17]. Experiments with wild mice demonstrated that non-cytotoxic Tc17 cells in skin are activated by commensal bacteria, and promote wound healing instead of inflammation[33,34]. The immunomodulatory[33] and skin migratory signature, CD40L expression, as well as the potential to secrete cytokines, such as IL-13, IL-10, and IL-22 instead of cytolytic molecules, implies that the here identified CD8$^+$ helper

T cells exert rather regenerative functions at barrier sites, such as the skin[21]. All no-ncytotoxic CD8$^+$ helper T-cell subsets expressed partly shared, unique TCR repertoires suggesting that antigens distinct from those activating conventional cytotoxic CD8$^+$ T cells are targeted. Helper CD8$^+$ T cells expressed CCR8 that can be induced during skin immunization[35,36]. CCR8$^+$ and CCR8$^-$ CD8$^+$ skin T$_{RM}$ have recently been shown to display distinct TCR repertoires in analogy to circulating CD8$^+$ helper T cells[37]. In murine experimental models, it was demonstrated that during skin immune responses, secondary T$_{RM}$ cells can be formed from precursors recruited from the circulation[38,39]. In addition, CD69$^+$CD103$^+$ CD4$^+$ T$_{RM}$ cells entering the circulation downregulate CD69, while continuously expressing CLA and partly CD103, analogous to our helper CD8$^+$ memory T cells[40]. Moreover, the obtained gene expression signatures of peripheral CD8$^+$ T helper subsets strongly resembled those of skin T$_{RM}$. We therefore hypothesize that significant numbers of CD8$^+$ helper T cells in blood represent circulatory variant of non-cytotoxic CD8$^+$ tissue-resident memory T cells.

Apart from their putative role in the maintenance of tissue homeostasis and integrity at barrier sites, circulating memory CD8$^+$ helper T cells are most likely also involved in the pathogenesis of autoinflammatory and allergic disorders. For example, Tc17 cells are critical for the migration and accumulation of pathogenic CD4$^+$ T cells into the central nervous system (CNS), and IL-17-producing CD8$^+$ T cells can be detected in active lesions in the CNS of MS patients[41–43]. IL-13-producing CD8$^+$ T cells have been associated with the induction and progress of asthma, as well as contact allergy dermatitis[44–47]. Finally, we, and others, have demonstrated a contribution of Tc17/Tc22 T cells in progressing psoriasis[48,49]. Helper CD8$^+$ T cells may induce the secretion of a distinct repertoire of chemokines by their adjacent cells, such as CCL20 from keratinocytes, which is associated with progressing

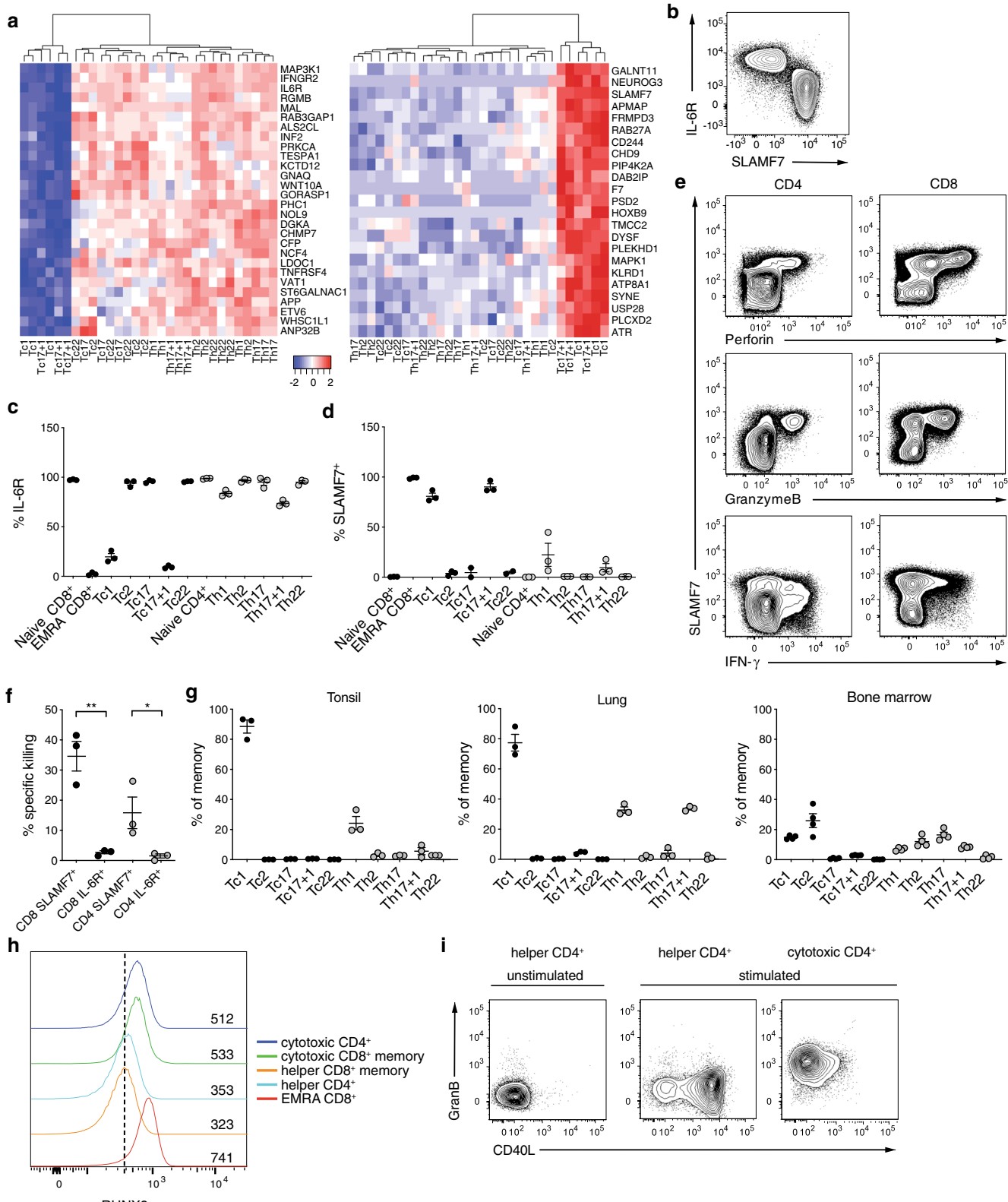

psoriasis. Furthermore, their capability to exert strong CD40L expression upon activation could be a critical additional functional driver of autoinflammatory processes. Targeting circulatory helper CD8+ T cells might offer a perspective to reduce systemic effects of autoimmunity and allergic disorders, such as autoantibody formation or the migration of cells to induce inflammation at barrier sites (skin and lung), which needs to be further examined.

It remains to be assessed how helper CD8+ T cells are induced. The expression of the CD8+ helper T-cell markers CCR4 and CLA can be imprinted in T cells upon activation in skin-draining lymph nodes[50–52]. We hypothesize that a specific repertoire of antigens, presumably originating from the skin, is involved in this primary induction of non-cytotoxic CD8+ T cells. Their skin-homing signature suggests that they are able to migrate back to

**Fig. 6 SLAMF7 and IL-6R distinguish cytotoxic and helper-type T cells. a** Heatmaps of normalized and scaled genes distinguishing cytotoxic (Tc1 and Tc17 + 1) and noncytotoxic CD8$^+$ and CD4$^+$ T-cell subsets (Tc2, Tc17, Tc22, Th1, Th2, Th17, Th17 + 1, and Th22). **b** Representative dot plot of IL-6R and SLAMF7 expression of human lymphocytes. **c, d** Frequency of IL-6R and SLAMF7 among human naive CD45RA$^+$CCR7$^+$, EMRA CD45RA$^+$CCR7$^-$, and memory CD45RA$^-$CD8$^+$ (black circles) and CD4$^+$ (gray circles) T-cell subsets. Mean ± SEM. **e** Co-expression of SLAMF7 with granzyme B, perforin, and IFN-γ in polyclonally activated human CD8$^+$ and CD4$^+$ T cells. **f** Redirected killing of α-CD3-coated p815 mouse mastocytoma cell line after 6 h cocultivation with sorted CD45RA$^-$ CD4$^+$ or CD8$^+$ T cells expressing either SLAMF7 or IL-6R. Lysis is calculated by the reduction in the % of viable p815$^+$ cells in the presence of α-CD3 compared to uncoated controls ($n = 3$). Mean ± SEM. Student's t test. *$P < 0.05$, **$P < 0.01$, ***$P < 0.001$. **g** Abundance of different memory CD8$^+$ and CD4$^+$ T-cell subsets in human tonsils, lung, and bone marrow. Mean ± SEM. **h** Human EMRA CD8$^+$ (CD45RA$^+$CCR7$^-$), helper CD8$^+$ memory (CD45RA$^-$CXCR3$^-$CCR4$^+$), cytotoxic CD8$^+$ memory (CD45RA$^-$CXCR3$^+$CCR4$^-$), helper CD4$^+$ memory (CD45RA$^-$CD28$^+$), and cytotoxic CD4$^+$ memory (CD45RA$^-$CD28$^-$CD57$^+$) T cells were sorted and RUNX3 expression assessed. Numbers in plot indicate the MFI. **i** Granzyme B and CD40L co-expression of sorted and polyclonally activated human helper CD4$^+$ memory (CD45RA$^-$CD28$^+$) and cytotoxic CD4$^+$ (CD45RA$^-$CD28$^-$CD57$^+$) T cells compared to unstimulated control.

the origin of the initial challenge. We could demonstrate here that the CD8$^+$ T cell compartment of conventional laboratory mice held under SPF conditions, in contrast to immune competent pet shop mice, almost lacked helper, as well as cytotoxic CD8$^+$ T-cell subsets comparable to the human situation. These observations may support the findings of Belkaid and colleagues that identified Tc17 cells as responders against commensal-derived peptides[33,34]. Therefore, further studies have to be performed in adapted murine systems or in human model systems, in order to provide physiological environments that are apparently critical for the generation of CD8$^+$ helper T cells.

Our data supports the notion that T-cell memory embodies universal immunological codes independent of MHC-restriction, and associated CD4 versus CD8 expression. CD4$^+$ and CD8$^+$ T cells share the flexibility in their differentiation into different memory T cell subsets with unique analogous cytokine profiles and effector functions. Moreover, our and others' findings demonstrate that not only does the CD4$^+$ T cell compartment include cytotoxic effector cells, but also that a substantial fraction of memory CD8$^+$ T cells display functional helper characteristics. Circulating CD8$^+$ helper T cells may represent a distinct force in cellular immunity, being able to migrate to and exert specific functions at barrier sites, such as the skin, and hence sustain a versatile immunity. Based on our results, the capabilities and functions of CD8$^+$ T cells in human health and disease have to be reevaluated.

## Methods

**Mice**. C57BL/6 J mice were purchased from Charles River Laboratories, bred and housed under SPF conditions at the institution's animal facility (Charité). Wild *Mus musculus domesticus* were purchased from pet shops in Berlin, Germany. Sex and age-matched mice were used for each experiment. All animal experiments were performed in accordance with German law and approved by the LaGeSo, Berlin.

**Human blood and tissue**. Human blood was obtained from buffy coats (DRK Blutspendedienst Ost) or from healthy volunteers. Blood and skin biopsies from psoriasis and atopic dermatitis patients, as well as skin biopsies from healthy volunteers were obtained by K. Wolk, R. Sabat, G. Kokolakis, and G. Heine, Charité, after informed consent (Supplementary Table 1). Disease activity status was classified by characteristic skin lesions and medical records. Atopic dermatitis was diagnosed according to published criteria[53]. All atopic dermatitis patients expressed increased serum IgE (median 756 kU/L, IQR [3987;174], norm < 100 kU/L) and were sensitized against >3 common aeroallergens. Tonsils were kindly provided by C. Romagnani, lung tissue from A. Hocke, and bone marrow from the Cell and Tissue Harvesting Core Unit (all Charité). All sampling and processing was approved by the Institutional Review board of the Charité.

**Cell isolation and cultivation**. *Murine tissues*. Spleens were mashed through a 70-micron cell strainer to generate single-cell suspensions and erylysis performed with ACK lysis buffer (Gibco). Cells were cultivated in RPMI 1640 medium supplemented with 100 U/mL penicillin, 0.1 mg/mL streptomycin, and 10% inactivated FCS (PAA).
*Human skin*. Subcutaneous fat was removed with a scalpel, the biopsy cut into small pieces and desintegrated with an automated gentleMACS dissociator (Miltenyi Biotec) followed by 12 h digestion in human medium containing 1 mg/mL collagenase IV (Thermo Fisher Scientific) and 25 U/mL Benzonase (Sigma) at 37 °C under continuous rotation. Remaining fragments were smashed on a 70 μM strainer to obtain single-cell suspensions.

*Human blood*. Peripheral blood mononuclear cells (PBMCs) were separated from heparinized whole blood by density centrifugation (Biocoll, Biochrom). Cells were cultured in RPMI 1640 medium (Gibco) supplemented with 100 U/mL penicillin, 0.1 mg/mL streptomycin, and 10% inactivated human AB serum (Pan Biotec).
*Human lung, tonsils, and bone marrow*. Lung and tonsils were cut into small pieces prior to mashing through a 70 μM filter with the end of a syringe plunger. Bone marrow was mashed directly. After washing, cells were passed through a 30 μM filter and target cells enriched by density centrifugation.
*P815 cell line*. The p815 mouse mastocytoma cell line was purchased from DSMZ and cultivated in RPMI 1640 medium supplemented with 100 U/mL penicillin, 0.1 mg/mL streptomycin, and 10% inactivated FCS (PAA) at 0.5–5 × 10^5 cells/mL.

**Flow cytometry**. Following fluorochrome-conjugated antibodies titrated for their optimal dilution were used for surface cell staining: Human: CD3 (clone UCHT1), CD4 (RPA-T4), CD8 (RPA-T8), CD45RA (HI100), CCR7 (G043H7), CCR6 (G034E3), CCR4 (L291H4), CCR10 (6588-5), CXCR3 (G025H7), SLAMF7 (162.1), IL6-R (UV4), CD57 (HCD57), CD28 (CD28.2), CD103 (Ber-ACT8), CD69 (FN50), CD101 (BB27), CD9 (HI9a). Mouse: CD3 (17A2), CD4 (RM4-5), CD8 (53-6.7), CD62L (MEL-14), CD44 (IM7), CD19 (6D5), NK1.1 (PK136), IL-6R (D771517), SLAMF7 (4G2). Surface staining was conducted for 15 min at room temperature (RT). To avoid unspecific Fc-receptor binding, human cells were stained in the presence of 1 mg/mL Beriglobin (CSL Behring) and murine cells with 2 μg/mL anti-FcγR (2.4G2). For intracellular stainings, dead cells were excluded by incubation with LIVE/DEAD dye (ThermoFisher Scientific) for 10 min prior to surface staining. Intracellular staining (30 min at RT) was carried out after fixation and permeabilization of cells with FACS-Lysing and FACS-Perm2 Solution (BD Biosciences) with the following antibodies: Human: granzyme B (GB11), perforin (B-D48), IL-4 (MP4-25D2), IL-17A (BL168), IL-22 (22URTI), IFN-γ (B27), CD40L (5C8). Mouse: IL-17 (TC11-18H10.1), IL-13 (eBio13A), CD40L (MR1). For intranuclear transcription factor staining Nuclear Factor Fix Perm (Biolegend), and the antibodies GATA3 (REA174) and AHR (FF3399) were used. Murine perforin (S16009A) was stained after FoxP3 fixation and permeabilization (Biolegend). Flow cytometric analysis was conducted on a LSRII flow cytometer (BD) and the FACS data were analyzed with FlowJo (Tree Star). All samples were pregated on alive, doublet-free lymphocytes. In mouse samples, CD19 and NK.1.1 were additional stained in the dump channel and excluded from analyses.

**Antigen-specific cells**. Per 10^7 cells, 5 μL Streptamers (CMV pp65 NLVPMVATV, EBV BMLF-1 GLCTLVAML; influenza M GILGFVFTL; all IBA Lifesciences) were incubated with 4 μL Strep-Tactin-PE (IBA) in a total volume of 50 μL over night at 4 °C. PBMC of HLA-A*0201 MHC-I-expressing donors were stained with antibody mixture for 10 min at RT followed by incubation with the subsequent Streptamer for 45 min at 4 °C, and analyzed by flow cytometry.

**Mass cytometry**. A total of 3 × 10^6 human PBMCs were stained with metal-conjugated antibodies in 100 μl final volume in a 96-deep well plate for 30 min at RT. A total of 0.5 μM cisplatin (Fluidigm) was added during the last 10 min to enable the exclusion of dead cells. Cells were washed twice with flow cytometry buffer (0.1% BSA, 2 mM EDTA in PBS) and pellet was resuspended in 300 μl of 2% paraformaldehyde in PBS, and incubated at 4 °C over night. The following day, fixed cells were washed once with flow cytometry buffer and permeabilized using 1× saponin permeabilization buffer (eBiosciences) in PBS on ice for 1 h. After washing with flow cytometry buffer, the pellet was resuspended in 1 ml of 12.5 nM nucleic acid Intercalator-Ir solution (Fluidigm), and incubated 30 min at RT. Cells were washed twice with flow cytometry buffer, twice with ultrapure water, and adjusted to 5 × 10^5 cells/mL for acquisition. Measurement was conducted on a CyTOF2 mass cytometer upgraded to Helios specifications (Fluidigm). EQ Four Element Calibration Beads (0.1×; Fluidigm) were added to the samples for data normalization of the FCS file using the CyTOF software. Tuning was performed

each day before measurement according to manufacturer's guidelines. Following antibodies were conjugated with metals (MaxPar, Fluidigm) in house, titrated for their optimal dilution, and used for CyTOF staining: CD45 (HI30), CD1c (L161), TCRγ/δ (B1), CD69 (FN50), CD11b (ICRF44), CCR4 (L291H4), CD20 (2H7), CD127 (A019D5), CD123 (6H6), CD103 (Ber-ACT8), CD14 (M5E2), CXCR3 (G025H7), CD11c (Bu15), CD28 (CD28.2), CD8 (GNM/134D7), IgD (IA6-2), CD56 (NCAM16.2), CD45RO (UCHL1), CCR6 (G034E3), CD3 (UCHT1), CCR7 (G043H7), CD4 (TT1), CD25 (2A3), SLAMF7 (162.1), CD38 (HIT2), CD39 (A1), CD49a (WM59), IL-6R (UV4), HLA-DR (L243), CD16 (3G8).

**Cell sorting**. For cell sorting, human $CD8^+$ and $CD4^+$ T cells were enriched from PBMCs with corresponding microbeads (Miltenyi Biotec), stained for the indicated surfaces markers, and subsets were sorted on a FACS Aria (BD). For the sorting and analysis of live cells, 100 nM DAPI (Sigma-Aldrich) or 1 mg/mL propidium iodide (ThermoFisher Scientific) was added.

**In vitro restimulation**. For the detection of the cytokine secretion potential, $1 \times 10^6$ cells/mL were stimulated polyclonally with 10 ng/mL phorbol-12-myristat-13-acetat (PMA) and 1 mg/mL PMA/ionomycin (Sigma-Aldrich) in the presence of 2 mg/mL BrefeldinA (Sigma-Aldrich). After stimulation for 6 h at 37 °C, 5% $CO_2$, the cytokines were stained intracellularly as indicated above and measured by flow cytometry.

**In vitro cultivation**. For long-term cultivation, 96-well plates were coated with 1 μg/mL α-CD3 (clone UCHT1, BD Bioscience) and 3 μg/mL α-CD28 (clone CD28.2, BD Bioscience) over night at 4 °C. A total of $5 \times 10^4$ cells/well were loaded in AB medium supplemented with recombinant cytokines IL-7/IL-15 or IL-2/IL-12 (all Miltenyi, 10 ng/mL final concentration). After 48 h, cells were transferred onto uncoated plates and cultivated for five further days. Upon restimulation with PMA/Iono in the presence of BrefA, the cells were stained intracellularly and measured by flow cytometry.

**Redirected lysis assay**. P815 cells were rested for 24 h in FCS-free medium prior to the experiment. The cells were labeled with 10 μM cell proliferation/viability dye (eBioscience) and cultivated 30 min at RT in the presence or absence of 5 μg/mL αCD3 (clone UCHT1, BD Bioscience). A total of $1 \times 10^5$ coated or uncoated p815 were co-cultured with sorted $CD45RA^-SLAMF7^+IL-6R^-$ or $CD45RA^-SLAMF7^-$ $IL-6R^+$ $CD8^+$ or $CD4^+$ T cells isolated from PBMC at E:T ratio 10:1 in 200 μL whole p815 medium in a 96-well plate. P815 incubated alone or with 0.05% Tween-20 (Sigma Aldrich) served as controls. After 6 h incubation at 37 °C, 5% $CO_2$, the cells were washed with PBS, stained for CD3 (clone SK7) and CD69, and fixated 20 min with 4% PFA (Sigma Aldrich) at 4 °C. The viability was assessed by the reduction of the viability dye intensity among viability dye$^+CD3^-$ cells. The percent of specific lysis was calculated as $100 \times [(\% \text{ alive } p815_{(E:T)} - \% \text{ alive } p815_{(E:T, \alpha CD3)})/\% \text{ alive } p815_{(E:T)}]$.

**Multiplex ELISA**. Sorted $CD8^+$ and $CD4^+$ T-cell subsets were cultivated at $1 \times 10^4$ cells/100 μL AB medium in αCD3/αCD28 (1 μg/mL, clone UCHT1/3 μg/mL, clone CD28.2, both BD) coated 96-well plates for 24 or 72 h. Multiplex ELISA (Q-Plex™ Array, Quansys) was performed with 50 μL supernatant according to the manufacturer's protocol. Infrared emission was analyzed with Odyssey (LiCor) at various intensities. Data analysis was performed with Q-view software (version 3.09, Quansys).

**Keratinocyte activation assay**. Supernatants from stimulated T cells, or medium as control, were transferred at 2% final concentration to primary human keratinocytes (CellSystems) cultured in KGM medium (Lonza), and incubated for a further 24 h. Cells were harvested and induction of CCL20 and CCL26 was assessed by RT-qPCR.

**RT-qPCR**. TRIzol® Reagent (ThemoFisher Scientific) was utilized for RNA isolation according to the manufacturers' protocol and reverse transcription of mRNA was conducted, as described previously[54]. Triplicates of each sample were analyzed by real-time PCR (StepOne plus, ThermoFisher scientific) using Maxima Probe/ROX qPCR Master Mix, ready-to-use detection assays for CCL20, CCL26, CCR4, and the house keeping gene HPRT containing double-labeled probes (all ThermoFisher Scientific). Gene expression was calculated relative to HPRT expression.

**RNA sequencing**. Total RNA from sorted Tc1/Th1, Tc2/Th2, Tc17/Th17, Tc17 + 1/Th17 + 1, and Tc22/Th22 T-cell subsets was isolated with Macherey and Nagel RNA Isolation kit. Quantity and quality were assessed with a Bioanalyzer 2100 device using the RNA 6000 Nano Kit (both from Agilent Technologies). Poly-(A)-selection was performed using the NEBNext Poly(A)mRNA Magnetic Isolation Module (NEB) according to the manufacturer's protocol. mRNA libraries were prepared with the NEBNext Ultra RNA Library Prep Kit for Illumina (NEB). All libraries were analyzed with the Agilent DNA 1000 Kit and quantified using the Qubit® dsDNA BR Assay Kit (ThermoFisher). After equimolar pooling, all samples

were sequenced on an Illumina HiSeq 1500 system with High Output chemistry v4 (50 cycles, single-read). Raw data were quality controlled by fastQC and reads were aligned to the GRCh37 (Ensembl) human genome using *bowtie2*[55]. Reads were summarized per gene using the featureCount algorithm implemented in the R-package *Rsubread*. Raw counts of protein-coding genes were normalized and transformed. The combat algorithm (package *sva*) was used to remove variances in gene expression which were associated with the CD4–CD8 contrast only and donor-specific differences[56]. Therefore, CD4 or CD8 samples were combined with the donor id and used as batch definition. Top 1000 most variable expressed genes across all samples of either non-compensated or compensated data were used to generate heatmaps of unsupervised hierarchical clustering (euclidean distances of scaled data and complete linkage). PCA: normalized and compensated data for all genes were scaled and subjected to a PCA using singular value decomposition. The first two principle components were shown in the plot. Signature genes distinguishing the cell subsets were identified by fitting generalized linear multinomial models to the normalized data via penalized maximum likelihood (package *glmnet*). RNA-Seq data for dermal and epidermal subsets were obtained from GEO database (GSE83637)[17]. Raw counts were combined with raw expression data from this study, normalized, and variance stabilized transformed (package *DESeq2*, version 1.14.1). In order to remove technical differences between the two studies, a batch compensation using the donor ids as batch definition was done (package *sva*). Top 1000 most variable expressed genes across all samples from this set were used in a PCA. PC1 and PC2 are displayed in the plot. Comparison of our dataset with published GEO sets that have analyzed Th-cell subsets GSE43005 (human, ex vivo), GSE49703 (human, after 36 h in vitro stimulation), and GSE14308 (mouse, in vitro generated) was conducted by an extraction of differentially expressed genes of the Th subpopulations of the GSE datasets (false discovery rate (fdr)-corrected, $P$ adj > 0.05, minimal absolute log2 fold change ≥ 1). After calculation of log2 fold changes of all Tc subpopulations of our dataset, the enrichment of the Th-signatures in the distribution of the log2 fold changes of the upregulated and of the downregulated genes of the Tc subpopulations were calculated (Kolmogornov–Smirnov test). ORA: differential expressed genes between Tc1/Tc17 + 1 cells as one group and Tc2/Tc17/Tc22 cells as the other group were determined by fitting models of negative binomial distributions to the normalized and log2-transformed data using the DESeq2 package in R[57]. Raw $P$ values were adjusted for multiple testing using fdr. Significant differential expressed genes were determined by adjusted $P$ values < 0.05 and a minimal absolute log2-fold change of 2. Overrepresentation of genes belonging to the biological process branch of the gene ontology system within the sets of either upregulated or downregulated genes were analyzed with the topGO package in R (topGO: Enrichment Analysis for Gene Ontology. R package version 2.26.0.). The background set were all genes in the analysis and the overrepresentation was calculated using the classical Fisher test. Due to the high redundancy of the gene ontology system, raw $P$ values were not compensated for multiple testing.

**TCR-sequencing**. Analyses of TCR repertoires was performed by next-generation sequencing of CDR3 β-chain[58,59]. Genomic DNA was isolated from FACS sorted Tc-cell subsets using AllPrep DNA/RNA Micro Kit (Qiagen) followed by CDR3-TCR-β locus amplification, using primers covering all functional Vβ- and Jβ-genes, and sequencing with the Illumina HiSeq System. The primary analysis of raw sequencing data, including subsequent clone grouping and clonotype generation was performed as previously described, using the free open-source clonotyping IMSEQ analytic platform[60]. Reads with an average quality score <30 were excluded from the analysis. Shared clonotypes were calculated using the Jaccard index calculating the percentage of number of shared clonotypes of two populations divided by the sum of unique clonotype numbers of the two populations with shared clonotypes excluded[22]. Shared reads were calculated as the average of the sum of read frequencies of the shared clonotypes of the selected populations. Gene allele frequencies have been extracted from the T-cell repertoires using Python 3.6 and plots, where created with Circos Genome Data Visualization software.

**Statistical analysis**. Data is presented as mean ± standard error of the mean (SEM) or mean ± standard deviation. All statistics were conducted with GraphPad Prism (GraphPad Software) or R statistical environment. The $n$ indicates the number of samples/donors. $P$ values were set as: $*P < 0.05$, $**P < 0.01$, $***P < 0.001$.

**Reporting summary**. Further information on research design is available in the Nature Research Reporting Summary linked to this article.

## Data availability

RNA-sequencing data are deposited and accessible under NCBI GEO database number GSE115103.

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

## Acknowledgements

The authors thank the BCRT Core Units Flow Cytometry, Cell and Tissue Harvesting and Next-Generation Sequencing for their professional help. L.L. was supported by the International Max Planck Research School for Infectious Disease and Immunology (IMPRS-IDI), Berlin and is a member of the ZIBI Graduate School Berlin. This work was supported by SFB Tr36 project A11, BCRT Key Project D2, and the Einstein Center for Regenerative Therapies Project "TiSSueHeLP". F.M. acknowledges funding from the EFRE.NRW program OsteoSys (EFRE-0800427; LS-1-1-019c). We thank C. Romagnani/D.Hernandez, A. Hocke/K.Hönzke, G. Kokolakis, and G. Heine for the blood and tissue samples, and C. Berek, B. Jack, U. Klein, and J. Braun for critical reading.

## Author contributions

Conceptualization: L.L. and A.T.; methodology: L.L., S.W., S.D., R. Stark, R. Sabat, and K.W.; Formal analysis: K.J. and F.M.; investigation: L.L., S.W., K.W., C.N., and B.K.; writing—original draft: L.L.; funding acquisition: M.F. and A.T.; resources: N.B., M.N., R. Sabat, and K.W.; and supervision: M.F. and A.T.

## Funding

## Competing interests

The authors declare no competing interests.
