## [Peer Review File · Nature Communications]

REVIEWER COMMENTS

Reviewer #1 (Remarks to the Author):

Loyal et al. present some observations on memory CD8+ T cells and draw parallels with memory CD4+ T cells. They describe Tc1, Tc2, Tc17, Tc17+1 and Tc22 subsets and show cytokine profiles similar to Th1, Th2, Th17, Th17+1 and Th22 respectively. They also look at chemokine receptors, master-regulator transcription factors and SLAMF7/IL-6R to characterize these helper memory CD8+ T cells. The data presented are interesting and carry merit. The scope of these findings needs to be better defined, and some loose ends need to be addressed, especially in light of the fact that the authors are urging readers to reconsider the current understanding of the functional roles of CD4/CD8 T cells.

Comments:

1. A lot of observations/parallels have been drawn between memory CD4 and memory CD8 subsets. Do these also hold true for total CD8+ T cells? In order to truly revisit the helper vs. cytotoxic paradigm, there ought to be broader applicability. Cytotoxic CD4+ T cells have been described before, yet the majority of CD4+ T cells continue to play a helper role.
2. The cytokine profiles of CD8+ T cells are quite low, in spite of using PMA+Iono stimulation. It is important to investigate if these memory helper CD8+ T cells would respond to real antigens. In other words, the fact that some subsets 'can' produce certain cytokines under non-specific stim, doesn't necessarily mean they routinely do that as part of their functional profile.
3. The transcription factors *tbx21*, *gata3*, *rosc*, etc. were measured using RNASeq. The PCA looks good, but some flow/CyTOF data showing the actual expression of transcription factors at the protein level would help strengthen the case.
4. Some representative frequencies of Tc subsets should be reported. It will help in trying to understand how impactful their role might be.
5. The described memory Tc subsets are stated to play a helper role. Are they actually functionally capable of providing B cell help similar to CD4 T cells? Are there CD8+ counterparts to conventional CD4 Tfh?
6. Any possible mechanisms behind the correlation of SLAMF7/IL-6R with helper/cytotoxic functions? Fig 3b uses total human lymphocytes. Is the expression of SLAMF7 and IL-6R almost mutually exclusive in most subsets, including non-T cells?
7. Fig3h shows a substantial number of cytotoxic CD4+ T cells. Can exact frequencies be reported? Usually, these are very rare... perhaps are there reports in the literature which would match the frequencies obtained in these experiments?
8. Fig1: Frequencies of Tc17 and Tc22 are very low. Was the staining background and/or gating clear enough to actually define these subsets?
9. Fig1: Th17+1 have a slightly different profile from Th17+1 with little IL-17 or related cytokines. Are these functionally distinct from Tc1, since they predominantly produce IFN γ ? They show expression of *tbx21* and *rosc* in the RNA seq data, but do they express T-bet and ROR γ t? If yes, does this translate to any other functional differences between Tc1 and Tc17+1?
10. Fig4: We see some differences in Tc subsets between healthy and psoriatic patients. This is interesting, and further causal/necessary/sufficient type evidence would be needed to establish that memory helper CD8+ T cells play a role in psoriasis.

Reviewer #2 (Remarks to the Author):

Loyal et al. show in a surprisingly clear way that phenotypic attributes that are known to separate CD4 T cell subsets can also be found among CD8 T cells. They show this using unbiased RNAsequencing and multicolor flow analysis. Overall this is a very elegant study summarized in a succinct and clearly written manuscript.

The subset idea of CD8 T cells has been around for a long time but the data by Loyal et al. show similarities that go much beyond. Moreover, they reveal that SLAMF7 and IL-6R can be used to distinguish cells with cytotoxic (Slamf7+) from CD8 T cells that rather have a helper-type-like profile (IL-6R+). Moreover, the TCRb usage suggests limited antigen overlap and developmental independence, which strongly re-enforces the view that these are distinct types of cells. Also of particular interest is that the largest difference between cytotoxic (Tc1 and Tc17+1) from the helper subset (Tc2, Tc17, and Tc22) lies in the expression of genes associated with migration to peripheral tissues like skin.

One could ask for single cell resolved RNAsequencing data and if the described subsets can also be found among antigen-specific T cells during or after an infection. However, given the strong limitations in performing experimental work right now, I think the MS is already strong enough and deserves to be published. I have only a few minor points.

Minor points:

I assume the MS was initially drafted for a shorter format. I think some more information could be edited for nature communication. For example - why was it necessary to use SPF and pet shop mice? The authors could also provide a brief introduction to lesser known things like Th17+1/Tc17+1, AHR. Maybe expand a little more about the significance of these findings in the discussion?

Figure 2a: According to the short methods section, some batch corrections were made for the RNAseq data. What has been done should be better highlighted in the figure legend and explained in more detail in the methods section.

Figure 2b: I might be better to put circles around the populations and label the populations directly in the graph. Right now there are many colors and it is difficult to connect the information.

Figure 2g: Shows mice obtained from a Pet Store but there is no callout to these mice in the main text.

Figure 3c: Some color coding to highlight regions with larger overlap would be helpful.

Extended figure 1b: It would be informative to also show flow data of cytokine expression of Th subsets.

Reviewer #3 (Remarks to the Author):

SLAMF7 and IL-6R define distinct cytotoxic versus helper memory CD8+ T cells

In this manuscript, Loyal and colleagues have applied the standard methodology for separating CD4+ Th subsets onto the CD8+ T cell population, and in the process report a number of minor CD8+ T cell populations that have transcriptional, phenotypic, and some functional characteristics distinct from classical cytotoxic CD8+ T cells. Namely, the authors report on the presence of "Tc2", "Tc17", "Tc22", and Tc17+1" cells which variously produce IL-4/IL-13, IL-17A, IL-22, and a combination of IFN γ /IL-17A, respectively. These skewed cytokine production profiles broadly line-up with phenotypic and transcriptional traits of the corresponding well-described Th subsets. Finally, the authors put forward the markers SLAMF7 and IL-6R to separate the more classical Tc1 cells (+ Tc17+1 cells) from these non-canonical Tc2, Tc17, and Tc22 populations. TCR sequencing analysis supports the notion that separation of these populations based on these markers accurately identifies distinct developmental lineages of CD8+ T cells.

Overall, the manuscript marks a very bold attempt to reorient the field's classification of CD8+ T cell biology. However, throughout the paper, there appear to be two conclusions that the authors are trying to draw: (1) CD8+ T cells can be divided into subsets that are comparable to the known subsets of CD4+ T cells (i.e. Th1, Th2, Th17 and Th22); (2) CD8+ T cells can be divided into helper-type and cytotoxic subsets, with CD8+ helper cells similar to CD4+ helper cells. At times, the authors seem to pick and choose data to support one or the other conclusion and the two conclusions are not clearly discussed together. For example, the heatmap in Figure 2a supports the conclusion of highly similar subsets of CD4+ and CD8+ T cells, while the heatmap in Extended Figure 2a suggests more of a helper vs. cytotoxic division within the CD8+ T cell compartment. A clearer more cohesive and internally consistent model would be of substantial benefit to the reader.

Finally, a number of controls are missing (discussed in detail below), which we feel are critical to support the authors claims that the division of labor between CD8+ and CD4+ T cells should be reconsidered.

Major comments

The Tc17 and Tc22 populations comprise a very small fraction of the CD45RA- CD8+ T cell population. As it is known that the CD8+ T cell population contains a small fraction of MHC II-restricted T cells, which can be expanded experimentally (e.g. Cd4-/- mice; Tyznik et al., J Exp Med, 2004), the authors should provide evidence that these cells are classical MHC Class I-restricted T cells, as this is not guaranteed simply by the expression of CD8. This is of particular concern as the data in Figure 4 suggests a largely non-overlapping TCR repertoire between these cell populations and the "classical" Tc1 cells.

The paper lacks functional data to clarify how similar the IL-6R+ CD8+ T cell populations are to their purported CD4+ T cell match. For example, the authors fail to show that the helper-like CD8+ T cells can provide helper functionality *in vitro* or *in vivo*. Moreover, it is unclear whether the "helper" CD8+ T cell subsets also have cytotoxic functionality.

The authors have sorted CD8+ T cell populations based on their differential expression of chemokine receptors. However, they have failed to exclude innate-like T cell populations, confounding their conclusions with regard to the subset composition of conventional CD8+ T cells. For example, MAIT cells comprise on average 8% of total CD8+ T cells (Gherardin et al., Immunol Cell Biol, 2018) and based on their expression of chemokine receptors, likely comprise a substantial proportion of the Th17+1 population analysed by the authors. This hypothesis is supported by the data in Extended Data Figure 5, which shows a high fraction of TRBV6-1 and TRBV20-1 reads within the Tc17+1 populations. Use of TRBV6-1 and TRBV20-1 is enriched within the MAIT cell population (Lepore et al., Nat Commun, 2014). Additionally, the authors should confirm no gdT cells are contaminating their CD8+ T cell populations, which may skew their results.

By sorting CD8+ T cells using the same chemokine receptors that are used for the isolation of CD4+ T cell subsets, the authors excluded the large CD8+ TEMRA (CD45RA+CCR7-) population. What fraction of CD8+ memory T cells do TEMRAs comprise and what is their subset composition? How would inclusion of this population alter the reported fractional distribution of different Tc subsets?

Statistics have not been included in Figure 1c or 1d. This is particularly important in Figure 1d, as Tc2, Tc17 and Tc22 cells appear to produce similar levels of several cytokines. For example, production of IL-4 and IL-13 does not appear to differ between Tc2, Tc17 and Tc22 populations, suggesting that the functional classification of these cells is incorrect. Given that Tc2 and Tc17/Tc22 populations are distinguished based only on their expression of CCR6, it seems likely that these populations comprise a mixture of cells with different functional profiles.

According to the methods section, the RNA-seq data shown in Figure 2a has been batch corrected for both donor and cell type. Cell type is a biological condition and should not be batch corrected. In contrast, the authors have not batch corrected the data shown in Extended Figure 2a. The authors should explain the reason for batch correcting the data in Figure 2a but not in Extended Figure 2a. In Extended Figure 2a, samples from the same cell type appear to group together (in the absence of batch correction), suggesting that batch correction may not be necessary for this data. For reviewing purposes, it would be useful to see the original PCA plot prior to batch correction in order to establish whether batch correction is warranted.

In Figure 2e, the authors should include the results of their gene ontology overrepresentation analysis (and the method that they used for this – what software? Exactly which cell populations were compared?) and/or could include a pathway enrichment analysis, so that the statistical enrichment of particular gene pathways can be assessed. Additionally, looking at the PDF file provided, the heatmap appears to have been cut and pasted together and is misaligned in several places.

The mouse work included in this paper does not provide a meaningful contribution to the results/conclusions and should be removed.

Statistics should be added to Figure 2f so that it is clear which differences are statistically significant. For example, the authors suggest that Tc1 and Tc17+1 cells express the highest levels of Perforin and Granzyme B, but is the expression of these molecules statistically significantly increased in the Tc17+1 population relative to Tc2/Tc17/Tc22 cells?

The genes with increased and decreased expression in Tc1 and Tc17+1 cells relative to Tc2, Tc17 and Tc22 cells shown in Figure 2e, are largely absent from the lists of differentially expressed genes shown in Figure 3a. For example, granzymes do not show increased expression in the cytotoxic CD8+ T cell subsets in Figure 3a. What is the explanation for this inconsistency between Figure 2e and 3a? Have the data in Figure 2e been batch corrected? How was the glmnet analysis performed? The authors should provide further details.

In Figure 4d, the authors report the average TCR read overlap between the various Tc subsets. However, as the populations are sometimes of very different sizes (e.g. Tc1 cells are much more numerous than all other subsets), reporting average overlap does not provide an accurate picture of these data. For example: if 1% of Tc1 cells share a TCR with Tc2 cells, but 50% of the Tc2 cells share a TCR with Tc1 cells, this would be reported as only 25.5% overlap. However, from these raw data, one would conclude that the Tc2 population is substantially-derived from a common precursor as Tc1 cells. While such a conclusion would be much less obvious from the data reported in its current form. It would be preferable to report what fraction of each subset has a TCR found in each of the other subsets.

Minor comments

On line 77 and in later parts of the paper, the rationale for focussing on the tissue homing marker CLA and on the relationship between “helper” CD8+ T cells and tissue-resident memory T cells in the skin should be clarified.

Figure 3h – why are unstimulated cytotoxic CD4+ T cells not included?

Figure 3i should include statistics to indicate whether the increased expression of Runx3 in cytotoxic relative to helper populations is statistically significant. An FMO control should also be included for each of the populations to show that background fluorescence is equivalent.

Line 92-93 – the authors mention that NK, NKT and ILC1 cells expressed SLAMF7 but not IL-6R however these cell types are not shown in the associated plots (Extended Data Figure 3b,c).

Typo on line 99 – should be Tc17+1 not Tc1+1.

The conclusion the authors are trying to make in the paragraph starting at line 105 should be clarified. Are they suggesting a difference between CD4+ and CD8+ T cells in that single pathogen-specific CD4+ T cells can differentiate into the whole range of Th subsets, whilst CD8+ T cells either become helper (Tc2, Tc17, Tc22) or cytotoxic (Tc1, Tc17+1)?

What are the genes that are contributing to PC1 in Figure 4e? i.e. what is it that separates the “helper” CD8+ T cells and skin T cells from the Tc1/Tc17+1 cells and the CLA+/CLA- TEM cells?

To the referees

Subject:

22.07.2020

Nature communications submission NCOMMS-20-06285A

We would like to thank the referees for the time and effort spent on evaluating our work, the positive feedback as well as the critical but fair comments that helped us to significantly improve the manuscript. We hope that the enclosed point-by-point reply, together with the revised manuscript and figures, will clarify all perceived inconsistencies and alleviate all remaining concerns making our study now suitable for publication.

Referee #1:

“Loyal et al. present some observations on memory CD8+ T cells and draw parallels with memory CD4+ T cells. They describe Tc1, Tc2, Tc17, Tc17+1 and Tc22 subsets and show cytokine profiles similar to Th1, Th2, Th17, Th17+1 and Th22 respectively. They also look at chemokine receptors, master-regulator transcription factors and SLAMF7/IL-6R to characterize these helper memory CD8+ T cells. The data presented are interesting and carry merit. The scope of these findings needs to be better defined, and some loose ends need to be addressed, especially in light of the fact that the authors are urging readers to reconsider the current understanding of the functional roles of CD4/CD8 T cells.”

Specific comments:

1. “A lot of observations/parallels have been draw between memory CD4 and memory CD8 subsets. Do these also hold true for total CD8+ T cells? In order to truly revisit the helper vs. cytotoxic paradigm, there ought to be broader applicability. Cytotoxic CD4+ T cells have been described before, yet the majority of CD4+ T cells continue to play a helper role.”

We thank the reviewer for the opportunity to clarify this important perspective. The CD8+ T cell compartment, in analogy to CD4+ T cells can be dissected into naïve, memory and effector T cells. Naïve CD4+ and CD8+ T cells both display an undifferentiated phenotype, whereas memory and effector cells arise upon pathogen encounter that requires them to adapt highly specific characteristics that includes helper versus cytotoxic features. While the majority of CD4+ T cells display helper characteristics, a small fraction of approximately 2%

of the CD4⁺ T cells gain a cytotoxic phenotype (expression of lytic molecules, IFN γ , killing capacity) and have been demonstrated to play a role in anti-viral responses (Reviewed in Juno et al., Front. Immunol., 2017). Vice versa, most CD8⁺ T cells are cytotoxic (Tc1 and Tc17+1 memory, effector CD8⁺ T cells). However the fraction of Tc2, Tc17 and Tc22 accounting for approximately 3-5% of total CD8⁺ T cells display CD4⁺ helper like characteristics including the lack of cytotoxicity and the expression of CD40L – a central molecule to provide B cell help/APC licensing. We here do not question the dominating function of CD4⁺ versus CD8⁺ T cells but demonstrate that the flexibility within the T cell compartment goes beyond the classical division into CD4⁺ = helper and CD8⁺ = cytotoxic T cells.

2. “The cytokine profiles of CD8⁺ T cells are quite low, in spite of using PMA/Iono stimulation. It is important to investigate if these memory helper CD8⁺ T cells would respond to real antigens. In other words, the fact that some subsets ‘can’ produce certain cytokines under non-specific stim doesn’t necessarily mean they routinely do that as part of their functional profile.”

We utilized polyclonal activation (PMA/Iono for the flow cytometric data, α CD3/ α CD28 for the ELISA) in order to identify cytokine production capability in an antigen independent manner. While anti-viral responses are widely described and established for Tc1/Tc17+1 cells, the antigens of the helper subsets Tc2, Tc17 and Tc22 are unknown. CD8⁺ T cells producing Tc2/Tc17/Tc22 associated cytokines were described in several autoimmune disorders as discussed. Accordingly, we show an increase of the frequencies of these subsets in the PBMC of psoriasis patients as well as their resemblance in the transcriptome signature to skin CD8⁺ T cells. To strengthen the diverse functionality of those two groups of memory CD8⁺ T cells, we performed a directed lytic assay, additionally demonstrating the absence of killing capacities in the IL-6R expressing helper CD8⁺ T cell fraction (new Fig. 6f). In the overall, the identification of the helper CD8⁺ T cell antigens is of high interest and should be addressed in follow up studies, however, would exceed the scope of this manuscript.

*3. “The transcription factors *tbx21*, *gata3*, *rorc*, etc. were measured using RNASeq. The PCA looks good, but some flow/CyTOF data showing the actual expression of transcription factors at the protein level would help strengthen the case.”*

We agree with the reviewer, that it is important to demonstrate that the transcription factors are differentially expressed on protein level as well. While T-bet and RoR γ t expression in CD8⁺ T cells is widely established and published elsewhere (e.g. Chellappa et al., JImmunol., 2017), we show protein expression of the genes associated with the here characterized CD8⁺

helper subsets. We found GATA3 not only in Tc2 but also Tc17 and Tc22 and vice versa AHR expressed highest in Tc22 and Tc17 but to some extent also in Tc2 in RNA-Seq. We could verify those findings on the protein expression level as demonstrated in Fig. 2d.

4. “Some representative frequencies of Tc subsets should be reported. It will help in trying to understand how impactful their role might be.”

We show the frequencies of the different Tc subsets (compared to the different Th subsets) in Fig. 1b. Moreover, in Fig. 4c, the frequencies of the different Tc subsets in healthy donors and psoriasis patients are plotted.

5. “The described memory Tc subsets are stated to play a helper role. Are they actually functionally capable of providing B cell help similar to CD4 T cells? Are there CD8+ counterparts to conventional CD4 Tfh?”

We demonstrated in a previous publication that CD40L⁺ CD8⁺ T cells are capable to activate APC and induce antibody production by B cells (Frentsch et al., 2013, Blood). Moreover, we show now in the revised manuscript in the new Fig. 6f, that IL-6R⁺ “helper” CD8⁺ memory T cell subsets are not able to exert classical CD8⁺ T cell assigned killing functions.

The T_{FH} marker CXCR5 was no DEG in RNA-Seq. Flow staining of PBMC revealed all CXCR5⁺ memory CD8⁺ T cells are among CXCR3⁺CCR6⁻CCR4⁻ Tc1 cells accounting for 2-5% of total memory in line with the data from R. Ahmeds lab that all CXCR5⁺ CD8⁺ T cells in mouse co-express CXCR3 (see graph below). Those CXCR5⁺ cells migrate in response to CXCL13 into tonsil B cell follicles where we found solely Tc1 cells (new Fig. 6h). Since this data would not provide novel information, we did not include it into the paper.

Response Letter Fig. A: Co-expression of CXCR5 with CXCR3 and CCR6 in PBMC pre-gated on alive memory CD8⁺ T cells.

6. “Any possible mechanisms behind the correlation of SLAMF7/IL-6R with helper/cytotoxic functions? Fig 3b uses total human lymphocytes. Is the expression of SLAMF7 and IL-6R almost mutually exclusive in most subsets, including non-T cells?”

Fig. 6 b-e, Extended Data Fig. 7a, b, e, f and especially Extended Data Fig. 3a demonstrate that SLAMF7 is expressed on all lymphocytes associated with cytotoxicity including cytotoxic CD4⁺ T cells, NK cells, NKT cells, ILC1 cells, effector CD8⁺ T cells (CCR7⁻CD45RA⁺), Tc1 and Tc17+1 memory CD8⁺ T cells, g/d T cells. We extended the heatmap of Extended Data Fig. 3a in order to show SLAMF7 and IL-6R expression on the different (non)-lymphocyte subsets (see below). We observe no SLAMF7 or IL-6R expression in CD20⁺ B cells with exception of the small fraction of plasma B cells that do express both markers. CD16⁺ monocytes express IL-6R while CD14⁺ monocytes additionally display little SLAMF7 expression. Myeloid and plasmacytoid DCs are IL-6R⁺ SLAMF7⁺.

Response Letter Fig. B: Heatmap of of median marker intensities of indicated markers in the different PBMC derived subsets measured by Mass Cytometry.

We have now included some information regarding the potential function of SLAMF7 and IL-6R expression among T cells:

“Self-ligation of SLAMF7, previously described as a NK cell marker, was recently shown to increase cytotoxic degranulation and IFN- γ secretion capacity not only in NK cells but also in CD8⁺ T cells (Comte et al., 2017). We could demonstrate that anti-viral CD8⁺ T cells are mostly SLAMF7⁺. In contrast, non-cytotoxic T cells displayed IL-6R expression allowing classical IL-6 signaling, which has been associated with anti-inflammatory processes (Schaper and Rose-John, 2015).”

7. *“Fig3h shows a substantial number of cytotoxic CD4+ T cells. Can exact frequencies be reported? Usually, these are very rare... perhaps are there reports in the literature which would match the frequencies obtained in these experiments?”*

Around 2.2% of CD4⁺ T cells are Perforin expressing cytotoxic CD4⁺ T cells with outliers up to 35% in acute/chronic virus infection (Appay et al., J Immunol., 2002). These frequencies are in line with our findings and reflect the dot plot of Fig. 3e – now Fig. 6e. In contrast, Fig. 3h – now Fig. 6j shows the expression of GranzymeB versus CD40L in *sorted* helper versus cytotoxic CD4⁺ T cells with a purity of >97%.

8. *“Fig1: Frequencies of Tc17 and Tc22 are very low. Was the staining background and/or gating clear enough to actually define these subsets?”*

We show a representative dot plot of our gating strategy in Fig. 1a. Moreover, cytokine profiles in flow cytometry as well as ELISA from sorted subsets and the RNA-Seq demonstrate a specific pattern of transcription factor and cytokine expression/secretion (Fig. 1c, d, Fig. 2c,d, Extended Data Fig. 1c, d). To further assess the purity of this gating strategy scRNA-Seq should be performed in near future based on the novel, here presented information of CD8⁺ T cell heterogeneity. We agree that scRNAseq would provide further insights into the purity of the analysed cell subsets. However at this point this should be addressed in follow up studies, since it would exceed the scope of this manuscript.

9. *“Fig1: Th17+1 have a slightly different profile from Th17+1 with little IL-17 or related cytokines. Are these functionally distinct from Tc1, since they predominantly produce IFN γ ? They show expression of *tbx21* and *rorc* in the RNA seq data, but do they express *T-bet* and *ROR γ t*? If yes, does this translate to any other functional differences between Tc1 and Tc17+1?”*

Despite their striking similarity, Tc1 and Tc17+1 cells clearly differ from each other in the expression of several genes such as IL23R, IFNGR1, VCAM1, CD38, TIGIT. We have extended Fig. 3a for the according information in order to highlight how those two population differ from each other. Moreover, Fig. 2a and Extended Data Fig. 2a, b demonstrate that the difference between type 1 and type 17+1 is in analogy to the observation made from CD4⁺ T cells within our data set as well as in comparison to data published elsewhere.

10. *“Fig4: We see some differences in Tc subsets between healthy and psoriatic patients. This is interesting, and further causal/necessary/sufficient type evidence would be needed to establish that memory helper CD8+ T cells play a role in psoriasis.”*

We wish to refer here to the cited publications of Hijnen et al., 2013 and Res et al., 2010 that demonstrate an overrepresentation of IL-17 and IL-22 producing CD8⁺ T cells in psoriatic lesion. We additionally show that these IL-17 and IL-22 producing CD8⁺ T cells can be identified by chemokine receptors and induce CCL20 secretion by keratinocytes generating a chemoattractant gradient that further recruits such cells into the lesions in a chemokine receptor dependent manner.

.

.

.

Referee #2:

“Loyal at al. show in a surprisingly clear way that phenotypic attributes that are known to separate CD4 T cell subsets can also be found among CD8 T cells. They show this using unbiased RNAsequencing and multicolor flow analysis. Overall this is a very elegant study summarized in a succinct and clearly written manuscript.

The subset idea of CD8 T cells has been around for a long time but the data by Loyal at al. show similarities that go much beyond. Moreover, they reveal that SLAMF7 and IL-6R can be used to distinguish cells with cytotoxic (Slamf7+) from CD8 T cells that rather have a helper-type-like profile (IL-6R+). Moreover, the TCRb usage suggests limited antigen overlap and developmental independence, which strongly re-enforces the view that these are distinct types of cells. Also of particular interest is that the largest difference between cytotoxic (Tc1 and Tc17+1) from the helper subset (Tc2, Tc17, and Tc22) lies in the expression of genes associated with migration to peripheral tissues like skin.

One could ask for single cell resolved RNAsequencing data and if the described subsets can also be found among antigen-specific T cells during or after an infection. However, given the strong limitations in performing experimental work right now, I think the MS is already strong enough and deserves to be published. I have only a few minor points.”

We wish to thank the reviewer for this nice summary highlighting the importance of our manuscript as a connective piece of puzzle for several loose ends around T cell phenotypes and functions. We agree that single cell RNAseq analyses would be the logical next step to specify the heterogeneity of the here defined CD8⁺ memory T cell subset in a more precise manner and to potentially associate individual cellular functional potentials with distinct TCR clonotypes. This definitely should be assessed in the near future.

Specific comments:

1. “I assume the MS was initially drafted for a shorter format. I think some more information could be edited for nature communication. For example - why was it necessary to use SPF and pet shop mice? The authors could also provide a brief introduction to lesser known things like Th17+1/Tc17+1, AHR. Maybe expand a little more about the significance of these findings in the discussion?”

We thank the reviewer for this important suggestion. The manuscript was indeed initially submitted as letter version and now extensively overworked to provide more information. Specifically, addressing the terms mentioned above we added the following sentences with according references that provide further in depth information into the manuscript:

Usage of SPF and pet shop mice:

“We could demonstrate here that the CD8⁺ T cell compartment of conventional laboratory mice held under SPF conditions, in contrast to immune competent pet shop mice, almost lacked helper as well as cytotoxic CD8⁺ T-cell subsets comparable to the human situation. These observations may support the findings of Belkaid and colleagues that identified Tc17 cells as responders against commensal derived peptides^{33,34}. Therefore, further studies have to be

performed in adapted murine systems or in human model systems in order to provide physiological environments that are apparently critical for the generation of CD8⁺ helper T cells.”

Difference between Tc1 and Tc17+1 cells:

“Despite their striking similarity, Tc1 and Tc17+1 cells clearly differ from each other in the expression of several genes such as IL23R, IFNGR1, VCAM1, CD38, TIGIT and CCR1, while Tc2, Tc17 and Tc22 cells share different combinations of genes including PTGDR2 (CRTH2), ICOS, TNFSF11 (RANK) and ITGAE (CD103) (Fig. 3a).”

Role of AHR in T cell differentiation, function of Th17+1 cells:

“Combinations of the four chemokine receptors CCR4, CCR6, CCR10 and CXCR3 were utilized for the identification of Th1, Th17+1, Th2, Th17 and Th22 type CD4⁺ memory T-cell subsets⁵⁻⁸.”

2. *“Figure 2a: According to the short methods section, some batch corrections were made for the RNAseq data. What has been done should be better highlighted in the figure legend and explained in more detail in the methods section.”*

We wanted to analyze similarities of Tc and Th subsets in a non-supervised way. In order to do this, usually a number (e.g. 1000) of high variable genes across all samples will be selected and the data will be analyzed with methods of dimension reduction like the principle component analysis. Since this selection is unbiased, unwanted covariates with high variable expression patterns in the data set may overlay the biological groups of interest. We have two such covariates in our data set, the basic differences between CD4⁺ and CD8⁺ T cells and the donor heterogeneity. We have run the PCA without batch compensation and color the points according to the donors:

All samples from one donor (blue) are shifted along the second principle component (PC2). Then we have colored the same PCA according to CD4⁺ (orange) and CD8⁺ (blue) T cells:

The samples belonging to the CD8⁺ T cells are shifted to the right side of the first principle component, indicating a strong influence of the CD4⁺/CD8⁺ differences in this analysis. Taken together, these figures demonstrate that the two covariates strongly overlay the subset specific information of our interest. In order to remove the bias caused by the donor heterogeneity and by the basic CD4⁺/CD8⁺ differences, we decided to use a batch compensation algorithm (ComBat) implemented in the sva-package (Jeffrey T. Leek, W. Evan Johnson, Hilary S. Parker, Elana J. Fertig, Andrew E. Jaffe, Yuqing Zhang, John D. Storey and Leonardo Collado Torres (2020). sva: Surrogate Variable Analysis. R package version 3.22.0 in R. The algorithm was developed to remove batch effects using a parametric empirical Bayes framework. The basic methodology is described in detail in Johnson and Rabinovic (Adjusting Batch Effects in Microarray Expression Data Using Empirical Bayes Methods, Biostatistics 2007).

The data were normalized, log₂-transformed and genes with zero counts in all samples were removed prior to run the ComBat algorithm. We used a combination of donor and CD4/CD8 group definition, resulting in 6 batches (each donor with either CD4 or CD8) for the compensation. No information about the distinct subsets were used in the batch compensation. After the compensation donor and CD4/CD8 samples were equally distributed in the principle component analysis.

Now corresponding subsets of Th - and Tc- cells are clustering together apart from other subsets in the principle component analysis, indicating strong similarities in their gene expression profiles.

Please note, that the figure here is the result from the PCA of the 1000 top variable genes (according to the other figures above), whereas the PCA shown in the manuscript was done with all genes after compensation.

The Extended Data Fig. 2a (now Extended Data Fig. 2b) is showing the hierarchical clustering of the 1000 top variable expressed genes of the data set without compensation. In general, samples of the same subsets are clustering together, but due to the strong influence of the CD4⁺/CD8⁺ differences, Tc- (orange) and Th- (blue) samples can be found in separated clusters. The similarities between corresponding Th- and Tc subsets are overlaid by the CD4⁺/CD8⁺ differences. However, this heatmap demonstrates in an unsupervised way, that Tc1 and Tc17+1 cells cluster together, apart from the other TC subsets, which appear more similar to the Th subsets. This behavior can also be seen in the PCA with the uncompensated data above, where the Tc1 and Tc17+1 are on the left side at PC1.

Since our responses here will be attached to the publication of the manuscript, we refrained from further extending the elaborate Methods section of the manuscript for clarity.

3. "Figure 2b: I might be better to put circles around the populations and label the populations directly in the graph. Right now there are many colors and it is difficult to connect the information."

We thank the reviewer for this very good suggestion how to improve the visualization of the figure and changed it accordingly (see below).

4. *“Figure 2g: Shows mice obtained from a Pet Store but there is no callout to these mice in the main text.”*

Thank you for the important hint. We had split Fig. 2 into two new figures for better visualization and now mention pet shop mice in the main text referring to the according figure (now Fig. 3e).

5. *“Figure 3c: Some color coding to highlight regions with larger overlap would be helpful.”*

We assume the reviewer is here referring to Fig. 4c – now Fig. 5c. Since the overlaps are also related to the size of the different Tc subsets, we found it somewhat arbitrary to highlight certain overlaps. Instead, we utilized the Jaccard index to provide the overlap information in a size unbiased manner in the Fig. 5d, e.

6. *“Extended figure 1b: It would be informative to also show flow data of cytokine expression of Th subsets.”*

We show ELISA data of cytokines secreted by Th subsets (n = 3) side by side to the Tc subsets (n = 10) from two different time points (24hrs and 72hrs) in Fig. 1d and Extended Data Fig. 1d. Due to the higher sensitivity of the ELISA and the time spanning, we believe the informative value is rather given here.

Referee #3:

"In this manuscript, Loyal and colleagues have applied the standard methodology for separating CD4+ Th subsets onto the CD8+ T cell population, and in the process report a number of minor CD8+ T cell populations that have transcriptional, phenotypic, and some functional characteristics distinct from classical cytotoxic CD8+ T cells. Namely, the authors report on the presence of "Tc2", "Tc17", "Tc22", and Tc17+1" cells which variously produce IL-4/IL-13, IL-17A, IL-22, and a combination of IFN γ /IL-17A, respectively. These skewed cytokine production profiles broadly line-up with phenotypic and transcriptional traits of the corresponding well-described Th subsets. Finally, the authors put forward the markers SLAMF7 and IL-6R to separate the more classical Tc1 cells (+ Tc17+1 cells) from these non-canonical Tc2, Tc17, and Tc22 populations. TCR sequencing analysis supports the notion that separation of these populations based on these markers accurately identifies distinct developmental lineages of CD8+ T cells.

Overall, the manuscript marks a very bold attempt to reorient the field's classification of CD8+ T cell biology. However, throughout the paper, there appear to be two conclusions that the authors are trying to draw: (1) CD8+ T cells can be divided into subsets that are comparable to the known subsets of CD4+ T cells (i.e. Th1, Th2, Th17 and Th22); (2) CD8+ T cells can be divided into helper-type and cytotoxic subsets, with CD8+ helper cells similar to CD4+ helper cells. At times, the authors seem to pick and choose data to support one or the other conclusion and the two conclusions are not clearly discussed together. For example, the heatmap in Figure 2a supports the conclusion of highly similar subsets of CD4+ and CD8+ T cells, while the heatmap in Extended Figure 2a suggests more of a helper vs. cytotoxic division within the CD8+ T cell compartment. A clearer more cohesive and internally consistent model would be of substantial benefit to the reader.

Finally, a number of controls are missing (discussed in detail below), which we feel are critical to support the authors claims that the division of labor between CD8+ and CD4+ T cells should be reconsidered."

We wish to thank the reviewer for this elaborate summary and the critical reading as well as suggestions below. We overworked the manuscript substantially in order to gain a more cohesive structure and hope we could address all open issues satisfactorily.

Major comments:

1. *"The Tc17 and Tc22 populations comprise a very small fraction of the CD45RA- CD8+ T cell population. As it is known that the CD8+ T cell population contains a small fraction of MHC II-restricted T cells, which can be expanded experimentally (e.g. Cd4-/- mice; Tyznik et al., J Exp Med, 2004), the authors should provide evidence that these cells are classical MHC Class I-restricted T cells, as this is not guaranteed simply by the expression of CD8. This is of particular concern as the data in Figure 4 suggests a largely non-overlapping TCR repertoire between these cell populations and the "classical" Tc1 cells."*

This is an extremely important point, which we had asked ourselves in the past and had looked into. Notably, we have to separate the mouse model(s) relying on gene knockouts as mentioned here from physiological/human data. The increase of MHCII-restricted CD8+ T

cells CD4^{-/-} and similarly MHCII^{-/-} mouse models as described by Tzynik et al. and others are a result of deregulated T cell maturation in the thymus. CD4 co-receptor expression is required to stabilize CD4 lineage and phenotype during intrathymic maturation upon MHCII dependent TCR signalling. If this is not occurring, they artificially gain CD8⁺ T cell characteristics while being MHCII specific (and would express CD4 co-receptor under physiological conditions).

Ranasinghe et al. (Immunity, 2016) however interestingly could identify MHCII restricted CD8⁺ T cells in human which were further characterized in Nyanhete et al. (Sci. Rep., 2019). However, both publications report MHCII-restricted CD8⁺ T cells are characterized by a conventional CD8⁺ T cell associated phenotype including IFN- γ secretion and lytic capacities that excludes the possibility those are the here described helper CD8⁺ T cells.

2. "The paper lacks functional data to clarify how similar the IL-6R⁺ CD8⁺ T cell populations are to their purported CD4⁺ T cell match. For example, the authors fail to show that the helper-like CD8⁺ T cells can provide helper functionality in vitro or in vivo. Moreover, it is unclear whether the "helper" CD8⁺ T cell subsets also have cytotoxic functionality."

We agree that the functionality of the populations is a very important aspect as well. We have shown in a previous publication that CD40L expressing CD8⁺ T cells, which are as we show here, IL-6R⁺, can activate antibody production by B cells and provide APC licensing (Frentsch et al., Blood, 2013). In order to address the cytotoxicity related killing capacity we moreover performed a redirected lysis assay. We sorted SLAMF7⁺CD8⁺, IL-6R⁺CD8⁺ and SLAMF7⁺CD4⁺, IL-6R⁺CD4⁺ T cells respectively and co-cultured them with α CD3 coated P815 mouse mastocytoma cells. After 6hrs we assessed the amount of killed cells shown in the novel Fig. 6f, clearly demonstrating the killing capacity is limited to the SLAMF7⁺ fraction and absent in IL-6R⁺ cells of both T cell populations.

3. "The authors have sorted CD8⁺ T cell populations based on their differential expression of chemokine receptors. However, they have failed to exclude innate-like T cell populations, confounding their conclusions with regard to the subset composition of conventional CD8⁺ T cells. For example, MAIT cells comprise on average 8% of total CD8⁺ T cells (Gherardin et al., Immunol Cell Biol, 2018) and based on their expression of chemokine receptors, likely comprise a substantial proportion of the Th17+1 population analysed by the authors. This hypothesis is supported by the data in Extended Data Figure 5, which shows a high fraction of TRBV6-1 and TRBV20-1 reads within the Tc17+1 populations. Use of TRBV6-1 and TRBV20-1 is enriched within the MAIT cell population (Lepore et al., Nat Commun, 2014). Additionally, the authors should confirm no gdT cells are contaminating their CD8⁺ T cell populations, which may skew their results."

Again, the reviewer addresses a very important point by mentioning the innate-like T cell populations, especially MAIT cells. For a better visualization of the inter-subsets and inter-donor differences of the expressed TCR, we changed the display to Circos plot and included the J β chain usage (Extended Data Fig. 6). Additionally, we stained PBMC for chemokine receptors as well as the MAIT markers V α 7.2⁺ and CD161 and found that 1-4% of the memory CD8⁺ T cells express the combination of V α 7.2⁺ and CD161. Those cells were homogeneously CCR6⁺, did not express CCR4 and in parts dim levels of CXCR3. Accordingly, at least the CXCR3 positive fraction of these cells might overlap with the here characterized Tc17+1 subset. When utilizing chemokine receptor gating of the different Tc subsets, we found a fraction of 3-10% of Tc17+1 cells expressing V α 7.2⁺ and CD161. Interestingly, CD8⁺ MAIT cells share phenotypical properties with the type 17+1 CD3⁺ T cells (Dias et al., 2018, PNAS); therefore, the details of the relationship of those different subsets should be addressed in near future. We do now discuss this overlap MAIT cells with Tc17+1 cells in the manuscript. However, the presence of some MAIT cells within the Tc17+1 fraction does not impact central messages of our manuscript.

$\gamma\delta$ T cells can account for 0.3-3% of memory CD8⁺ T cells in peripheral blood (Garcillan et al., Front. Immunol., 2015). We assessed TCR $\gamma\delta$ expression of the different Tc subsets of 3 donors and found <3% $\gamma\delta$ T cells in all subsets. Altogether, gating out the $\gamma\delta$ T cells would be

4. "By sorting CD8+ T cells using the same chemokine receptors that are used for the isolation of CD4+ T cell subsets, the authors excluded the large CD8+ TEMRA (CD45RA+CCR7-) population. What fraction of CD8+ memory T cells do TEMRAs comprise and what is their subset composition? How would inclusion of this population alter the reported fractional distribution of different Tc subsets?"

We thank the reviewer for this remark. In general, there are multiple studies describing the heterogeneity of T_{EMRA} e.g. Koch et al. (Immun. Ageing, 2008), suggesting a further subdiversification into 3 subsets depending on CD27 and CD28 expression. The focus of our study was to assess whether the CD45RA⁻ memory CD8⁺ T cell compartment displays a CD45RA⁻ memory CD4⁺ T cell analogous spectrum and heterogeneity (as described by Sallusto and colleagues), whether memory CD8⁺ T cell subsets utilize the CD4⁺ T cell analogous or different differentiation mechanisms and how "helper CD8⁺" and "cytotoxic CD4⁺" T cells fit into this concept. We did not detect CD40L, CCR4, IL-6R expression (Frentsch et al., Blood, 2013, Extended Data Fig. 1a, Fig. 6c) or other signatures of "helper" CD8⁺ T cells within T_{EMRA}. Moreover, by TCR sequencing of purified T_{EM}, T_{EMRA} and T_{RM} from human blood, spleen, LN, BM and lung, Farber et al. recently demonstrated that T_{EMRA} do not possess any overlap in their TCR sequences with T_{EM} and T_{RM}, further highlighting that they probably consist of a

unique pool of chronically activated T cells (unpublished, shown at: Academy Colloquium T-cell memory: thinking outside the blood, Amsterdam). We therefore excluded T_{EMRA} from the in-depth analyses in our study.

Despite the fact, that CD8⁺ T_{EMRA} homogenously express SLAMF7, the expression of CXCR3 and CCR10 on fractions of the CD8⁺ T_{EMRA} (see Extended Data Fig.1a) suggests that also T_{EMRA}s are characterized by heterogeneous migration capacities. We performed selected flow cytometry analyses in order to elucidate whether there are correlations with further markers described in our publication (shown below). CXCR3⁺ CD8⁺ T_{EMRA} have a partially more terminally differentiated phenotype (CD57⁺) but do not differ from CXCR3⁻ in their exhaustion status (PD-1⁺). They may be more prone to migrate into skin (CLA⁺) and display a strong inverse correlation with the expression of lytic molecules Granzyme B and Perforin. This might be indicative of a migration versus effector function (killing) switch mechanism but requires further analyses that would exceed the focus of this paper.

Response Letter Figure C: Expression of the indicated markers CLA, PD-1, CD57, Granzyme B and Perforin on pregated, CXCR3⁺ and CXCR3⁻ CD45RA⁺CCR7⁻ CD8⁺ T_{EMRA} in human peripheral blood assessed by flow cytometry.

5. *“Statistics have not been included in Figure 1c or 1d. This is particularly important in Figure 1d, as Tc2, Tc17 and Tc22 cells appear to produce similar levels of several cytokines. For example, production of IL-4 and IL-13 does not appear to differ between Tc2, Tc17 and Tc22 populations, suggesting that the functional classification of these cells is incorrect. Given that Tc2 and Tc17/Tc22 populations are distinguished based only on their expression of CCR6, it seems likely that these populations comprise a mixture of cells with different functional profiles.”*

We utilized chemokine receptor staining – which is widely used to distinguish different CD4⁺ memory T cell subsets and checked the usability among CD8⁺ memory T cells. The cytokine profile we found in the different CD8⁺ T cell subsets is highly similar to the different CD4⁺ T cell subsets – also in terms of “producing similar levels of several cytokines”. The major difference of the subsets (in CD8⁺ and CD4⁺) can be found in the RNA-Seq data. Altogether, we here demonstrate for the first time in a systematic manner, that there are strong parallels between CD4⁺ and CD8⁺ T cells and the classical view of cytotoxic CD8⁺ and helper CD4⁺ has its limitations. We agree that the next step should be to perform single cell RNA-Seq in order to make sure that the complete diversity of T cell memory is unrevealed but this would be beyond the scope of this manuscript.

6. *“According to the methods section, the RNA-seq data shown in Figure 2a has been batch corrected for both donor and cell type. Cell type is a biological condition and should not be batch corrected. In contrast, the authors have not batch corrected the data shown in Extended Figure 2a. The authors should explain the reason for batch correcting the data in Figure 2a but not in Extended Figure 2a. In Extended Figure 2a, samples from the same cell type appear to group together (in the absence of batch correction), suggesting that batch correction may not be necessary for this data. For reviewing purposes, it would be useful to see the original PCA plot prior to batch correction in order to establish whether batch correction is warranted.”*

We wanted to analyze similarities of Tc and Th subsets in a non-supervised way. In order to do this, usually a number (e.g. 1000) of high variable genes across all samples will be selected and the data will be analyzed with methods of dimension reduction like the principle component analysis. Since this selection is unbiased, unwanted covariates with high variable expression patterns in the data set may overlay the biological groups of interest. We have two such covariates in our data set, the basic differences between CD4⁺ and CD8⁺ T cells and the donor heterogeneity. We have run the PCA without batch compensation and color the points according to the donors:

All samples from one donor (blue) are shifted along the second principle component (PC2). Then we have colored the same PCA according to CD4⁺ (orange) and CD8⁺ (blue) T cells:

The samples belonging to the CD8⁺ T cells are shifted to the right side of the first principle component, indicating a strong influence of the CD4⁺/CD8⁺ differences in this analysis. Taken together, these figures demonstrate that the two covariates strongly overlay the subset specific information of our interest. In order to remove the bias caused by the donor heterogeneity and by the basic CD4⁺/CD8⁺ differences, we decided to use a batch compensation algorithm (ComBat) implemented in the sva-package (Jeffrey T. Leek, W. Evan Johnson, Hilary S. Parker, Elana J. Fertig, Andrew E. Jaffe, Yuqing Zhang, John D.

Storey and Leonardo Collado Torres (2020). sva: Surrogate Variable Analysis. R package version 3.22.0) in R. The algorithm was developed to remove batch effects using a parametric empirical Bayes framework. The basic methodology is described in detail in Johnson and Rabinovic (Adjusting Batch Effects in Microarray Expression Data Using Empirical Bayes Methods, Biostatistics 2007).

The data were normalized, log2-transformed and genes with zero counts in all samples were removed prior to run the ComBat algorithm. We used a combination of donor and CD4/CD8 group definition, resulting in 6 batches (each donor with either CD4 or CD8) for the compensation. No information about the distinct subsets were used in the batch compensation. After the compensation donor and CD4/CD8 samples were equally distributed in the principle component analysis.

Now corresponding subsets of Th - and Tc- cells are clustering together apart from other subsets in the principle component analysis, indicating strong similarities in their gene expression profiles.

Please note, that the figure here is the result from the PCA of the 1000 top variable genes (according to the other figures above), whereas the PCA shown in the manuscript was done with all genes after compensation.

The Figure Extended Data Fig. 2a (now Extended Data Fig. 2b is showing the hierarchical clustering of the 1000 top variable expressed genes of the data set without compensation. In general, samples of the same subsets are clustering together, but due to the strong influence

of the CD4⁺/CD8⁺ differences, Tc- (orange) and Th- (blue) samples can be found in separated clusters. The similarities between corresponding Th- and Tc subsets are overlaid by the CD4⁺/CD8⁺ differences. However, this heatmap demonstrates in an unsupervised way, that Tc1 and Tc17+1 cells cluster together, apart from the other TC subsets, which appear more similar to the Th subsets. This behavior can also be seen in the PCA with the uncompensated data above, where the Tc1 and Tc17+1 are on the left side at PC1.

7. “In Figure 2e, the authors should include the results of their gene ontology overrepresentation analysis (and the method that they used for this – what software? Exactly which cell populations were compared?) and/or could include a pathway enrichment analysis, so that the statistical enrichment of particular gene pathways can be assessed. Additionally, looking at the PDF file provided, the heatmap appears to have been cut and pasted together and is misaligned in several places.”

We apologize for this lack of information and included the following information into the manuscript:

“Overrepresentation analysis (ORA): Differential expressed genes between Tc1/Tc17+1 cells as one group and Tc2/Tc17/Tc22 cells as the other group were determined by fitting models

of negative binomial distributions to the normalized and log₂-transformed data using the DESeq2 package in R (Love, M.I., Huber, W., Anders, S. Moderated estimation of fold change and dispersion for RNA-seq data with DESeq2. *Genome Biology* 15(12):550 (2014)). Raw p-values were adjusted for multiple testing using False Discovery Rate (fdr). Significant differential expressed genes were determined by adjusted p-values below 0.05 and a minimal absolute log₂-foldchange of 2. Overrepresentation of genes belonging to the biological process branch of the gene ontology system within the sets of either up- or down-regulated genes were analyzed with the topGO package in R (topGO: Enrichment Analysis for Gene Ontology. R package version 2.26.0.). The background set were all genes in the analysis and the overrepresentation was calculated using the classical Fisher test. Due to the high redundancy of the gene ontology system, raw p-values were not compensated for multiple testing.”

Moreover, we display the top 10 hits as identified by ORA in Extended Data Fig. 2c now:

Response Letter Figure D: Top 10 gene ontology terms from overrepresentation analysis (ORA) of genes upregulated in Tc1/Tc17+1 or Tc2/Tc17/Tc22 cells respectively.

8. *“The mouse work included in this paper does not provide a meaningful contribution to the results/conclusions and should be removed.”*

Since mouse models are widely used to address many aspects of T cell phenotype and function, we found it important to understand whether and to what extent our findings can be transferred into mouse models. We demonstrate here, that SPF housing conditions impair the development of the CD40L⁺ CD8⁺ T cells with helper characteristics as well as proper cytolytic compartment (SLAMF7⁺ expression, Perforin co-expression) (Fig.3e, Extended Data 7e, f) supporting the notion that memory CD8⁺ T cell responses should be analysed in mice bred under dirty, immune-challenging conditions in line with previous reports (Japp et al., Cytometry A, 2017; Rosshart et al., Science, 2019).

More importantly, the antigen(s) of helper CD8⁺ T cells are still unknown. The work of Yasmine Belkaid and colleagues suggest an important role of helper CD8⁺ T cells in the detection of commensal bacteria – a condition that is not sufficiently reflected under SPF housing conditions. Our data here underlines the importance to include helper CD8⁺ T cell subsets in the analysis of CD8⁺ T cell function(s) and analysing anti-viral responses in SPF mice will miss the contribution of those cells in the immune responses.

9. *“Statistics should be added to Figure 2f so that it is clear which differences are statistically significant. For example, the authors suggest that Tc1 and Tc17+1 cells express the highest levels of Perforin and Granzyme B, but is the expression of these molecules statistically significantly increased in the Tc17+1 population relative to Tc2/Tc17/Tc22 cells?”*

Since Perforin expression is dependent on recent activation of the cell we believe that adding statistics to the graph would be misleading. One can clearly see that at least one of the donors is having GranzymeB and Perforin⁺ cells among Tc17+1. Besides RNA-Seq results as well as SLAMF7 and Runx3 expression (Fig. 3a, 6d, e, i) highlight the potential killing capacity of these cells as demonstrated by directed lysis assay in Fig. 6f.

10. *“The genes with increased and decreased expression in Tc1 and Tc17+1 cells relative to Tc2, Tc17 and Tc22 cells shown in Figure 2e, are largely absent from the lists of differentially expressed genes shown in Figure 3a. For example, granzymes do not show increased expression in the cytotoxic CD8+ T cell subsets in Figure 3a. What is the explanation for this inconsistency between Figure 2e and 3a? Have the data in Figure 2e been batch corrected? How was the glmnet analysis performed? The authors should provide further details.”*

Fig. 2e (now 3a) and Fig. 3a (now 6a) are the result of different types of analysis with a different focus. For each of the analysis normalized and log2- transformed data without batch compensation were used. The genes shown in Fig. 2e (now 3a) are a selection of

differentially expressed genes, which are associated with specific biological processes. The genes in Fig. 3a (now 6a) derived from a glmnet approach. The glmnet algorithm is implemented in the R-package "glmnet" (Jerome Friedman, Trevor Hastie, Robert Tibshirani (2010). Regularization Paths for Generalized Linear Models via Coordinate Descent. Journal of Statistical Software, 33(1), 1-22. URL <http://www.jstatsoft.org/v33/i01/>.). This function fits generalized linear model via penalized maximum likelihood to the data. This algorithm is not a simple method to define differentially expressed genes, but to identify strong marker genes. Marker genes should be equally expressed in the samples within each biological group and should have strong expression differences between different biological groups. In contrast, the usual methods to define differentially expressed genes between groups allow more variation within the groups and smaller differences between the groups. To demonstrate the differences in the approaches, we have added the genes from 2e (now 3a) to the signatures from 3a (now 6a):

Genes from Fig. 2e (now 3a) are marked in red at the left side. It is obvious for most of these genes, that the expression patterns are more heterogeneous within the groups and less clear different between the two groups. TNFSRF4 is included in both figures (2e (now 3a) and 3a (now 6a)) and that is why it appears twice in the left figure. It is further important to note, that the results of each method depends also from the thresholds used in the analysis, allowing more strict or more relaxed results.

11. *"In Figure 4d, the authors report the average TCR read overlap between the various Tc subsets. However, as the populations are sometimes of very different sizes (e.g. Tc1 cells are much more numerous than all other subsets), reporting average overlap does not provide an accurate picture of these data. For example: if 1% of Tc1 cells share a TCR with Tc2 cells, but 50% of the Tc2 cells share a TCR with Tc1 cells, this would be reported as only 25.5% overlap. However, from these raw data, one would conclude that the Tc2 population is substantially-derived from a common pre-cursor as Tc1 cells. While such a conclusion would be much less obvious from the data reported in its current form. It would be preferable to report what fraction of each subset has a TCR found in each of the other subsets."*

The referee is addressing an important issue of correct TCR overlap plotting of different populations with various sizes. We apologize for the unclarity how the values of Fig. 5d were generated. We had utilized the Jaccard index (see also Becattini et al., Science, 2015) in order to plot shared (and not average) clonotypes and reads as explained the Methods section:

"Shared clonotypes were calculated using the Jaccard index calculating the percentage of number of shared clonotypes of two populations was divided by the sum of unique clonotype numbers of the two populations with shared clonotypes excluded. Shared reads were calculated as the average of the sum of read frequencies of the shared clonotypes of the selected populations."

We changed the figure legend of Fig. 5d accordingly for a better understanding.

Minor comments:

12. *"On line 77 and in later parts of the paper, the rationale for focussing on the tissue homing marker CLA and on the relationship between "helper" CD8⁺ T cells and tissue-resident memory T cells in the skin should be clarified."*

We thank the reviewer for this important comment and changed the manuscript accordingly. Firstly, we separated Fig.2 into two new figures for a better understanding and Extended Data Fig. 3a for more genes. As stated in the text: *"Gene ontology overrepresentation analysis (ORA) displayed a high enrichment of immunoregulatory genes and genes that are indicative of tissue migration in helper-type Tc2, Tc17 and Tc22 cells..."*. This includes the skin homing chemokine receptors CCR4 and CCR10, which were used for Tc subset gating and CCR8 as shown in Fig. 3a. As most prominent skin homing marker, we moreover assessed the protein expression of CLA in in the different Tc subsets in the following Fig.3c. Consequently, we compared the transcriptome of the circulatory helper subsets with skin located Trm in Fig. 4a and found helper CD8⁺ T cells rather resemble skin Trm than the classical, cytotoxic, circulatory CD8⁺ T cells in their gene expression signatures.

13. *“Figure 3h – why are unstimulated cytotoxic CD4+ T cells not included?”*

The central message of Fig. 3h – now Fig. 6j is the assess ability to express CD40L upon polyclonal activation. Cytotoxic CD4⁺ T cells – in contrast to helper CD4⁺ T cells - fail to express CD40L even after activation. Since CD40L expression is transient and highly stimulation dependent, showing unstimulated cytotoxic CD4⁺ T cells would not add extra informational content.

14. *“Figure 3i should include statistics to indicate whether the increased expression of Runx3 in cytotoxic relative to helper populations is statistically significant. An FMO control should also be included for each of the populations to show that background fluorescence is equivalent.”*

Fig. 3i – now 6i display flow cytometric data of Runx3 and the according MFI values of indicated, sorted subsets. We apologize for not having performed FMO controls in these experiments. However, the experiments shown are representative for 3 individual experiments, displaying always similar differences in RUNX3 MFI for helper CD8⁺ and CD4⁺ memory T cells versus cytotoxic CD4⁺ and cytotoxic CD8⁺ memory T cells. According to different experimental settings used in these experiments no statistics were performed.

15. *“Line 92-93 – the authors mention that NK, NKT and ILC1 cells expressed SLAMF7 but not IL-6R however these cell types are not shown in the associated plots (Extended Data Figure 3b,c).”*

We apologize for this lack of clarity. NK, NKT and ILC1 cells share the expression of CD56, which is shown in the Extended Data 7b. We added the information into the manuscript.

16. *“Typo on line 99 – should be Tc17+1 not Tc1+1.”*

Thank you very much for the correction, we changed it accordingly.

17. *“The conclusion the authors are trying to make in the paragraph starting at line 105 should be clarified. Are they suggesting a difference between CD4+ and CD8+ T cells in that single pathogen-specific CD4+ T cells can differentiate into the whole range of Th subsets, whilst CD8+ T cells either become helper (Tc2, Tc17, Tc22) or cytotoxic (Tc1, Tc17+1)?”*

This could be one interpretation of the data; the alternative one is that the priming sites are an important factor. The expression of skin homing receptors as well as skin T cell

resembling gene expression signature of helper CD8⁺ T cells indicate that they might be primed in skin draining lymph nodes in response to a set of pathogens/antigens that differs from classical viral infections that occur in lung/tonsil/etc. In the novel Fig. 6h we show that solely Tc1 cells but not helper CD8⁺ T cells are not found in lung and tonsils – which are typical sites of viral infections.

18. “What are the genes that are contributing to PC1 in Figure 4e? i.e. what is it that separates the “helper” CD8⁺ T cells and skin T cells from the Tc1/Tc17+1 cells and the CLA⁺/CLA⁻ TEM cells?”

In order to define the strongest candidate genes separating the two groups of cells, the data from the 1000 topvariable genes from the principle component analysis presented in the manuscript were used as input for the glmnet approach (Jerome Friedman, Trevor Hastie, Robert Tibshirani (2010). Regularization Paths for Generalized Linear Models via Coordinate Descent. Journal of Statistical Software, 33(1), 1-22. <http://www.jstatsoft.org/v33/i01/>.). The data of the selected genes were scaled and depicted in heatmaps.

Response Letter Figure E: Differentially expressed genes between indicated skin and blood derived CD8+ T cell subsets. Upper figure: genes upregulated in Tc1/Tc17+1/TEM. Lower figure: genes upregulated in “helper” CD8+ and skin CD8+ T cells.

REVIEWERS' COMMENTS:

Reviewer #2 (Remarks to the Author):

The authors have addressed all my points. This is an important study that deserves to be published.

REVIEWERS' COMMENTS:

Reviewer #2 (Remarks to the Author):

The authors have addressed all my points. This is an important study that deserves to be published. **We thank the reviewer for reading our revised manuscript and the support in publishing our manuscript.**